# An organ-on-chip model of pulmonary arterial hypertension identifies a BMPR2-SOX17-prostacyclin signalling axis

Alexander J. Ainscough[1,2], Timothy J. Smith[2], Maike Haensel[1], Christopher J. Rhodes[1], Adam Fellows [1], Harry J. Whitwell[3,4], Eleni Vasilaki[1], Kelly Gray[5], Adrian Freeman[5], Luke S. Howard [1], John Wharton [1], Benjamin Dunmore[6], Paul D. Upton [6], Martin R. Wilkins [1], Joshua B. Edel [2] & Beata Wojciak-Stothard [1✉]

Pulmonary arterial hypertension (PAH) is an unmet clinical need. The lack of models of human disease is a key obstacle to drug development. We present a biomimetic model of pulmonary arterial endothelial-smooth muscle cell interactions in PAH, combining natural and induced bone morphogenetic protein receptor 2 (BMPR2) dysfunction with hypoxia to induce smooth muscle activation and proliferation, which is responsive to drug treatment. BMPR2- and oxygenation-specific changes in endothelial and smooth muscle gene expression, consistent with observations made in genomic and biochemical studies of PAH, enable insights into underlying disease pathways and mechanisms of drug response. The model captures key changes in the pulmonary endothelial phenotype that are essential for the induction of SMC remodelling, including a BMPR2-SOX17-prostacyclin signalling axis and offers an easily accessible approach for researchers to study pulmonary vascular remodelling and advance drug development in PAH.

[1] National Heart and Lung Institute, Imperial College London, London, UK. [2] Department of Chemistry, Imperial College London, London, UK. [3] National Phenome Centre and Imperial Clinical Phenotyping Centre, Department of Metabolism, Digestion and Reproduction, Imperial College London, London, UK. [4] Section of Bioanalytical Chemistry, Division of Systems Medicine, Department of Metabolism, Digestion and Reproduction, Imperial College London, London, UK. [5] Emerging Innovations Unit, Discovery Sciences, BioPharmaceuticals R&D, AstraZeneca, Cambridge, UK. [6] Department of Medicine, University of Cambridge School of Clinical Medicine, Addenbrooke's and Royal Papworth Hospitals, Cambridge, UK. ✉email: b.wojciak-stothard@imperial.ac.uk

Pulmonary arterial hypertension (PAH) is a progressive disease of the pulmonary circulation characterised by the narrowing of pulmonary arteries and arterioles, leading to right ventricular hypertrophy and heart failure[1]. The disease is thought to be triggered by endothelial damage due to a combination of two or more hits, such as genetic alterations, hypoxia and inflammation, followed by exaggerated repair involving increased proliferation and resistance to apoptosis of cells in the arterial intimal and medial layers. In addition to this, endothelial-to-mesenchymal transition (EMT), aerobic glycolysis and conversion of the differentiated, contractile phenotype of smooth muscle cells to a proliferative, synthetic phenotype, are key contributors to vascular remodelling[2]. Existing drugs bring some symptomatic relief but, in most patients, do not arrest or reverse the disease[3].

Rare genetic mutations have been identified that increase susceptibility to PAH[4]. A key genetic change is a heterozygous germline mutation in the gene encoding the bone morphogenetic protein type II receptor (*BMPR2*). Studies report *BMPR2* mutations in over 80% of cases of familial PAH, and around 20% of sporadic or idiopathic PAH patients[5]. Non-genetic forms of PAH show reduced expression and increased degradation of the receptor[6]. While mutations in *BMPR2* and other TGF-β receptor superfamily member genes account for most cases of heritable PAH, other genes and genetic loci have also been linked with the condition[5,7].

Animal models do not fully reproduce the features of human PAH, which is a key obstacle to drug development[8,9]. Organs-on-chips have emerged in the last decade as a technology with huge potential to revolutionise in vitro disease modelling and increase the accuracy and throughput of pharmacological and toxicological screening. In addition, there is the potential to personalise medicine development through the use of patient-derived induced pluripotent stem cells and endothelial colony-forming cells (ECFCs).

Here, we present a microfluidic model of human pulmonary arterial endothelial–smooth muscle cell interactions cultured under the conditions of BMPR2 knockdown and hypoxia, the two known triggers of PAH. In this manuscript, this model is referred to as a two-hit model of PAH. We characterise functional and transcriptomic changes in these cells and validate the findings with the use of cells from PAH patients and mice with disabling *BMPR2* mutations and comparative analysis of transcriptomic data from human PAH. The study identifies a BMPR2–SOX17–prostacyclin signalling axis, essential for the regulation of pulmonary arterial smooth muscle cell proliferation and demonstrates therapeutic effects of the endothelin receptor antagonist Ambrisentan[10], a tyrosine kinase inhibitor Imatinib[11], and AZD5153, a novel inhibitor of bromodomain-containing protein 4 (BRD4)[12].

## Results

### Pulmonary artery-on-a-chip reconstitutes features of human lung arteries

The pulmonary artery-on-a-chip (PA-on-a-chip) was designed to monitor the responses of human pulmonary vascular endothelial and smooth muscle cells to pathological and potential therapeutic interventions. The polydimethylsiloxane (PDMS)-based device comprises two microfluidic channels (200 μm height × 1000 μm width) separated by a nanoporous polyethylene terephthalate (PET) membrane, with human pulmonary artery endothelial cells (HPAECs) and human pulmonary artery smooth muscle cells (HPASMCs) cultured on either side of the membrane (Fig. 1a, b). Membrane porosity (pore size 400 nm, density $2 \times 10^6$ pores/cm$^2$) enables direct communication between cells, reflective of the role of the basal lamina in pulmonary arteries[13,14]. Inlets and outlets of endothelial channels are connected with tubing and pulse dampeners, to allow circulation of culture medium, actuated by laminar shear stress under physiological flow rates of 6 dynes/cm$^2$, characteristic of human lung arterioles[15] (Fig. 1c, d and Supplementary Figs. 1–3). The basic principles of the design were derived from a lung-on-a-chip model[16].

The alignment of HPAECs and HPASMCs within the device was reminiscent of the arterial intimal and medial cell configurations in vivo (Fig. 1e–g and Supplementary Fig. 4). Flow-stimulated HPAECs showed enhanced junctional localisation of vascular endothelial (VE)-cadherin and significantly elevated expression of endothelial maturity markers PECAM-1 and KLF2 (Fig. 1h, i). Endothelial barrier functionality was evaluated by measuring the passage of fluorescently-labelled 40 kDa dextran, which has a Stokes radius similar to human plasma albumin[17], across the endothelial and smooth muscle layers stimulated with thrombin, a known vasoconstrictor and permeability factor[18]. HPAECs in the PA-on-a-chip showed enhanced endothelial barrier function and a larger increase in thrombin-induced permeability over baseline, compared with unstimulated controls (3.9-fold increase in PA-on-a-chip vs 1.5-fold increase in static culture) (Fig. 1j). HPASMCs co-culture did not affect endothelial barrier function under the study conditions (Supplementary Fig. 5).

### Establishment of a PAH model on-a-chip

Remodelled PAH vasculature is characterised by intimal and medial thickening (Fig. 2a, b). There is a significant need to understand how endothelial *BMPR2* haploinsufficiency, in combination with the second hit created by hypoxia or inflammation, elicits disease.

To recreate disease conditions in the PA-on-a-chip, HPAECs infected with AdBMPR2 shRNA were exposed to hypoxia (2% O$_2$) for 24 h. The adenoviral infection was optimised to achieve ~50% reduction in BMPR2 expression, reflective of BMPR2 haploinsufficiency in PAH (Fig. 2c).

BMPR2 knockdown and hypoxia, separately or in combination, did not affect endothelial permeability or proliferation in PA-on-a-chip (Fig. 2d and Supplementary Fig. 6a), in contrast with prior observations in static cell culture[19]. HPASMC proliferation was significantly increased under the two-hit conditions when BMPR2 knockdown and hypoxia were combined ($P < 0.01$; Fig. 2e). This response was inhibited by imatinib mesylate, a receptor tyrosine kinase inhibitor currently under investigation as an anti-proliferation treatment of PAH ($P < 0.05$; Fig. 2e). No significant loss of cells from endothelial or smooth muscle channels was noted and none of the treatments affected cell apoptosis (Supplementary Figs. 6b, c and 7).

### Genotype- and oxygenation-specific transcriptomic signatures from the two-hit microfluidic model of PAH

Control and BMPR2-deficient HPAECs co-cultured with HPASMCs under normoxic or hypoxic conditions underwent RNAseq analysis to identify key transcriptomic changes induced under the two-hit conditions. *BMPR2* knockdown alone induced differential expression of 1828 genes in HPAECs ($|\text{logFC}| > 0.25$, $P < 0.05$, comparison with controls), where, as expected, reduction in BMPR2 expression was the most significant change ($-2.9$-fold change, $p = 2.07 \times 10^{-9}$) (Fig. 2f). These genes showed enrichment in epithelial-to-mesenchymal transition (EMT), G2M checkpoint, myc targets, cell cycle (E2F targets) and DNA repair pathways (Hallmark, FDR < 0.01). HPASMCs co-cultured with these cells showed differential expression of 751 genes (Supplementary Fig. 8), associated predominantly with hypoxia, glycolysis, EMT and cholesterol homoeostasis (Hallmark, FDR < 0.01).

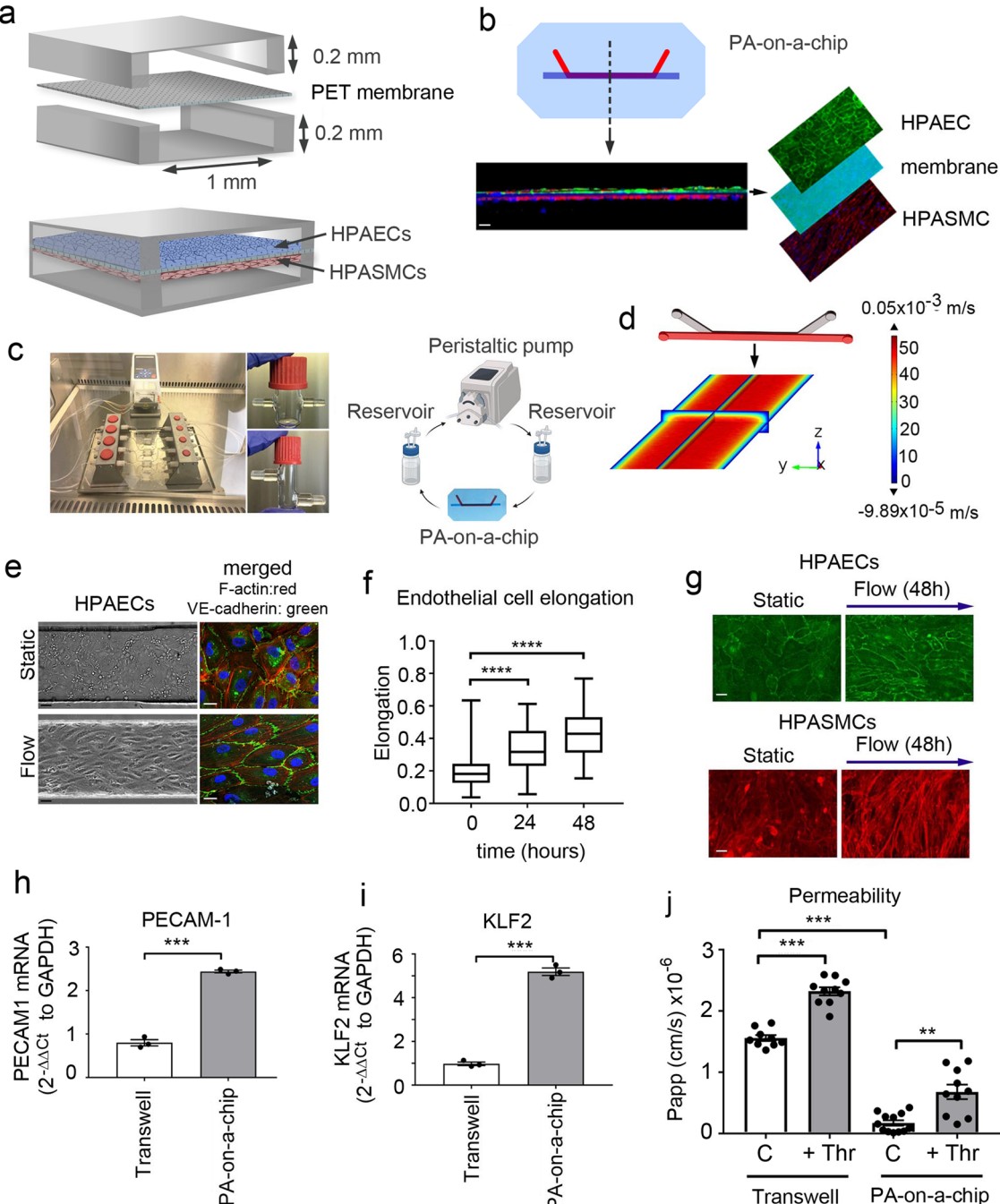

**Fig. 1 Pulmonary artery-on-a-chip reproduces the functionality of a blood vessel. a** Schematic diagram of the pulmonary artery-on-a-chip. **b** Z-section of endothelial-smooth muscle interface, fluorescent confocal imaging. HPAECs (green: VE-cadherin), HPASMCs (red: α-smooth muscle actin). Bar = 10 μm. **c** An image and a schematic diagram of the flow system set up comprising a peristaltic pump, media reservoirs and chips, perfused with culture medium at 6 dynes/cm². **d** Fluid–structure interaction (velocity profile) in the endothelial channel, COMSOL modelling. **e** Phase contract (left) and fluorescent microscopy (right) images of HPAECs grown in microfluidic channels under static and flow conditions, as indicated. F-actin: red, VE-cadherin: green. Bar = 10 μm. **f** Endothelial cell elongation; n = 4 individual chips per time point. Error bars indicate mean ± SEM of a one-way ANOVA with a Tukey's post-hoc correction test. ****P < 0.0001. **g** HPAEC and HPASMC phenotype under static conditions and underflow (48 h) in PA-on-a-chip; fluorescent microscopy, F-actin (red) and VE-cadherin (green); Bar = 10 μm. **h, i** mRNA expression of endothelial differentiation markers, PECAM-1 and KLF2 in cells treated, as indicated; qPCR. ***P < 0.001; Unpaired t-test. n = 3. **j** Effects of thrombin (1 U/mL) on endothelial barrier function in HPAECs co-cultured with HPASMCs in pulmonary artery-on-a-chip or in transwell dishes. Passage of FITC-dextran (1 mg/mL; 1 h) from top to bottom channel was used as a measure of endothelial permeability. Cells from three different biological donors grown 3–4 chips/transwells per treatment group, were used; n = 10–12. Error bars indicate mean ± SEM of a one-way ANOVA with a Tukey's post-hoc correction test. **P < 0.01; ****P < 0.0001, comparisons, as indicated.

The combination of BMPR2 knockdown with hypoxia-induced differential expression of 1090 genes in HPAECs and 895 genes in HPASMCs. The pattern of these changes is illustrated in volcano plots (Fig. 2g, h) and heatmaps showing distinct hypoxia-, genotype- and two-hit-related gene clustering (Supplementary Fig. 9). A list of differentially expressed genes (DEG) in HPAECs and HPASMCs in the two-hit model of PAH is provided in Supplementary Data 1.

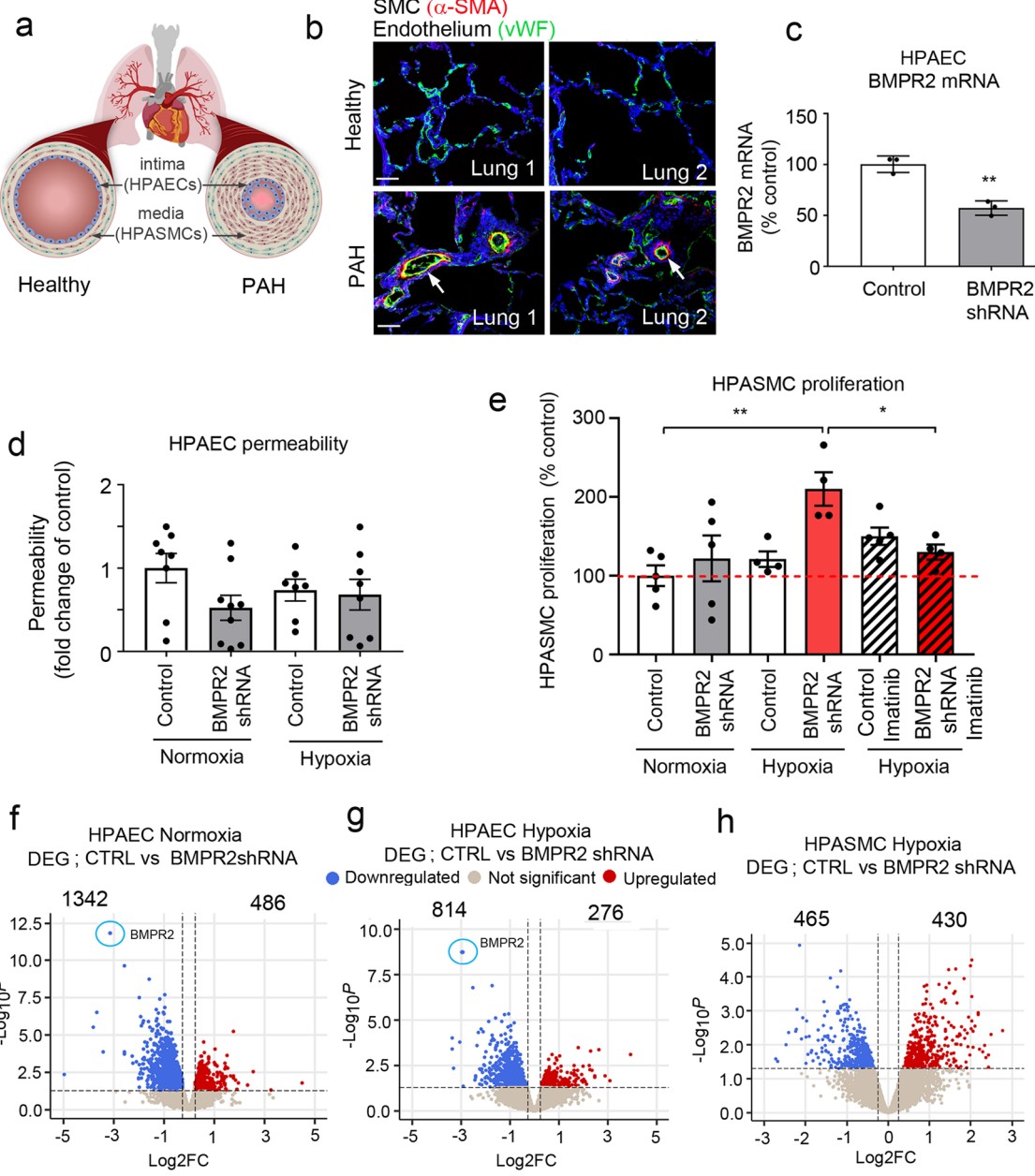

**Fig. 2 Two-hit model of PAH in pulmonary artery-on-a-chip. a** Schematic diagram of pulmonary vascular remodelling in PAH. **b** α-SMA (red) and vWF (green) staining in healthy and PAH lungs. Bar = 50 μm. **c** BMPR2 mRNA expression in HPAECs treated AdCTRLshRNA with Ad-BMPR2-shRNA-GFP. $n = 3$; **$P < 0.01$; unpaired $t$-test. **d** Effect of two-hit (hypoxia and BMPR2 knockdown, 24 h) on HPAEC permeability, measured as the passage of 40 kDa FITC-dextran (1 mg/mL) from top to bottom channel; $n = 9$ individual chips. **e** Effect of hypoxia and endothelial BMPR2 knockdown on HPASMC proliferation (24 h, EdU assay). Cells were untreated or treated with 10 μM Imatinib mesylate for 24 h. $n = 4$–5 individual chips. In **d**, **e** error bars are mean ± SEM; one-way ANOVA with Tukey post-hoc correction test. Volcano plots show differentially expressed genes (DEG) in **f** HPAECs treated with BMPR2shRNA, **g** HPAECs in the two-hit PAH model (BMPR2 shRNA and hypoxia), **h** HPASMCs in the two-hit PAH model. Downregulated genes are in blue and upregulated genes are in red; $P < 0.05$, 0.25-fold cut-off. $n = 4$.

Transcriptional profiling of HPAECs from the two-hit model identified multiple genes linked with known PAH pathways, including TGF-β signalling, NOS pathway, proliferation, angiogenesis, Notch signalling, EndoMT, ion channels, fibrosis, and inflammation (Fig. 3a and Supplementary Fig. 10a). ~80% of the observed changes could be attributed to *BMPR2* knockdown, including the downregulation of genes in TGF-β, NOS signalling pathway, junctional proteins and growth factor signalling mediators. DEG induced by hypoxia in BMPR2-deficient HPAECs and the corresponding GSEA pathway analysis data is provided in Supplementary Data 2 and 3.

HPASMCs in the two-hit PAH model showed associations with several pathways involved in vascular remodelling, including angiogenesis, apoptosis, inflammation, vasoconstriction and TGF-β signalling (Fig. 3b, Supplementary Fig. 10b). Only ~30% of these changes could be attributed to the endothelial *BMPR2* knockdown. The additional genes affected by hypoxic exposure were linked to glycolysis, adipogenesis, TNF-α, cell cycle, Rho signalling, and ECM interactions. DEG affected by HPAEC *BMPR2* silencing in normoxic HPAECs and HPASMCs are provided in Supplementary Data 4. Hypoxia-regulated HPASMC genes in the two-hit model and the corresponding GSEA pathway

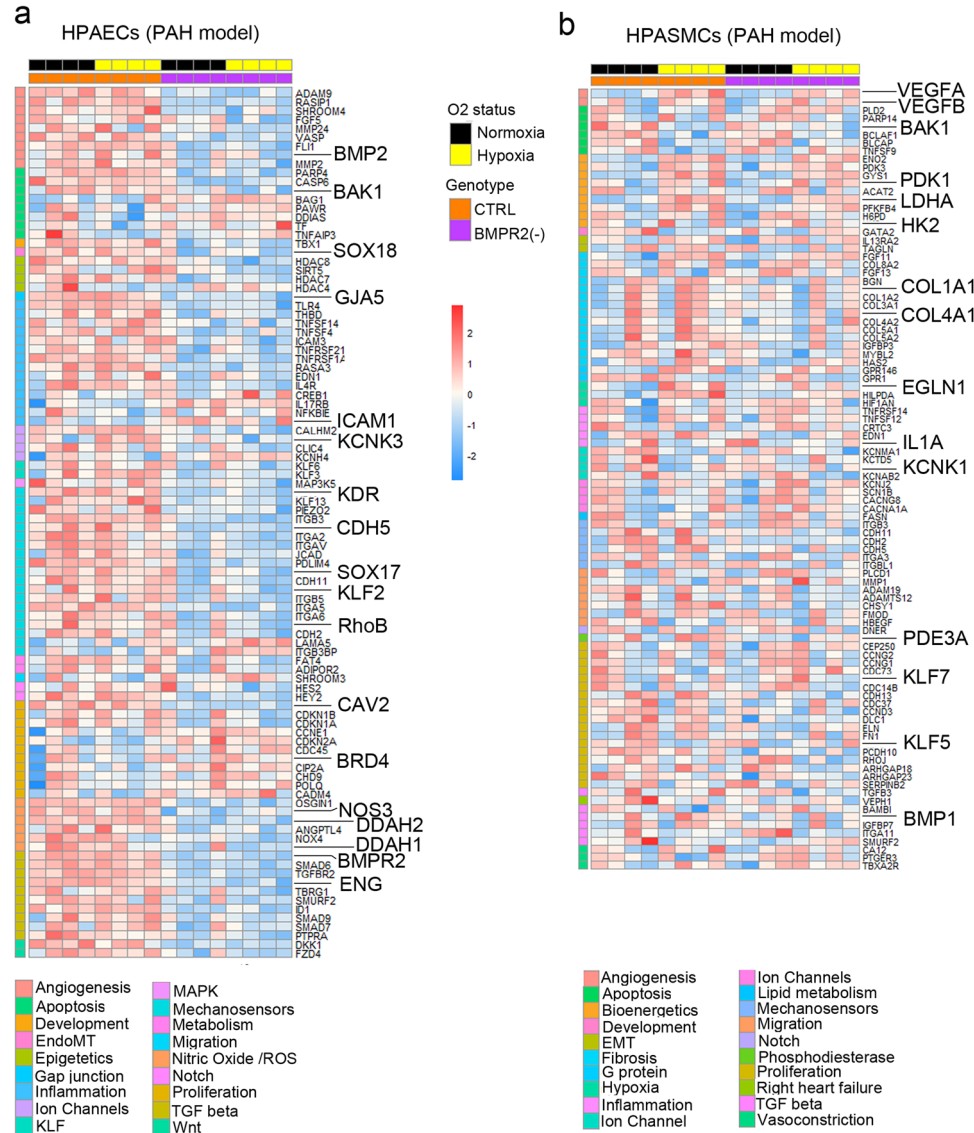

**Fig. 3 PAH pathways in the two-hit disease model.** Top DEG associated with cardiovascular diseases were grouped into categories and visualised as transcripts per million (TPM) heatmaps. Colour-coded pathways are shown underneath the heatmaps, with key gene symbols enlarged in **a** HPAECs and **b** HPASMCs. The genotype status (control and BMPR2 knockdown) and oxygen status (normoxia or hypoxia) are shown at the top of the heatmap in different colours, as indicated. Changes in gene expression are also colour coded, with blue denoting a lower relative gene expression and red denoting a higher relative gene expression. Each column represents 1 experimental repeat ($n = 4$/treatment group).

analysis data are provided in Supplementary Data 2 and 5, respectively.

Changes in the expression of selected HPAEC and HPASMC gene targets were validated by qPCR and immunofluorescent analysis (Supplementary Figs. 11 and 12). To further validate our findings, we measured the activity of hexokinase, the rate-limiting enzyme in the glycolysis and pentose cycle, in HPASMCs. Hexokinase activity in HPASMCs cultured under basal, control conditions was $0.86 \pm 0.043$ nM NADH/min/mL. It increased under hypoxic conditions ($3.4 \pm 0.9$-fold increase, $P < 0.05$, comparison with controls) and was further markedly elevated in HPASMCs cultured with HPAECs under the two-hit conditions ($6.3 \pm 0.4$-fold increase, $P < 0.0001$, comparison with controls) (Supplementary Fig. 13).

**Model validation with PAH cells with *BMPR2* mutations.** Blood-derived endothelial colony-forming cells (ECFCs) are often used as surrogates for pulmonary endothelial cells in PAH and display abnormalities in key pathways linked to the disease pathogenesis[20,21].

To investigate whether the effects seen in the two-hit model of PAH would yield similar findings in patient-derived cells, we opted to substitute HPAECs with ECFCs from PAH patients with disabling *BMPR2* mutations. The process of isolation and culture of ECFCs is illustrated in Fig. 4a. PAH ECFCs did not show major morphological differences with healthy controls (Fig. 4b, c and Supplementary Fig. 14). However, consistent with the responses seen in the two-hit PAH model, HPASMCs co-cultured with PAH ECFCs under hypoxic conditions showed a significant ~2-fold increase in cell proliferation (Fig. 4d). No changes in ECFC and HPASMC apoptosis were observed (Supplementary Fig. 15).

Normoxic PAH ECFCs with *BMPR2* mutations showed 1065 differentially expressed genes (DEG) versus healthy controls (Fig. 4e), associated predominantly with ROS, KRAS, mTOR signalling and adipogenesis (Hallmark, $p < 0.01$). 15% of this gene pool was shared with BMPR2-deficient HPAECs and included

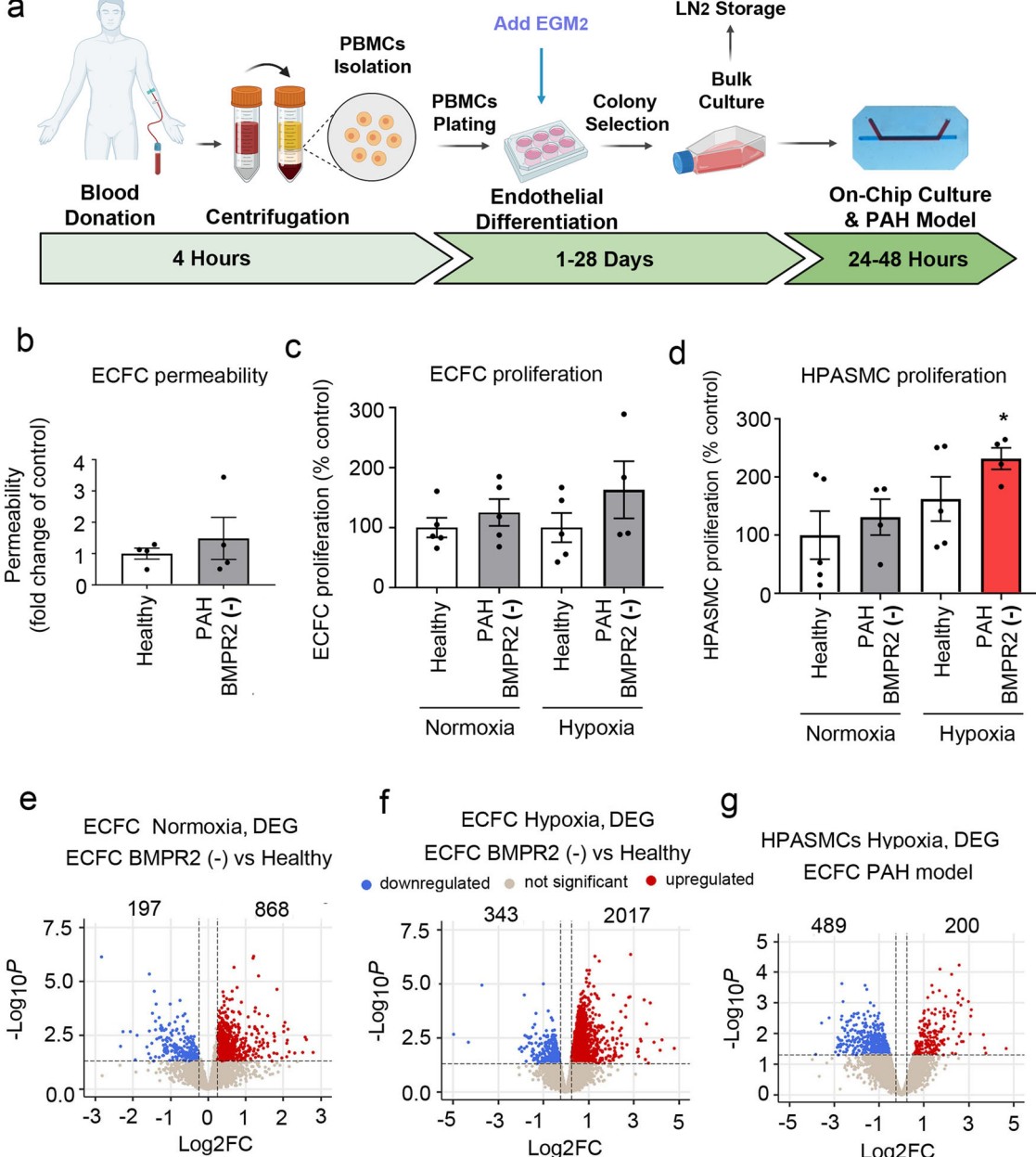

**Fig. 4 Model validation using cells from PAH patients with *BMPR2* mutations. a** Schematic diagram of isolation and culture of patient ECFCs. **b** Permeability of ECFCs from healthy individuals and PAH patients with BMPR2 mutations cultured in PA-on-a-chip. **c** Proliferation of healthy and PAH ECFCs in PA-on-a-chip under normoxic or hypoxic conditions (2% $O_2$, 24 h), as indicated. **d** Proliferation of HPASMCs co-cultured with control or patient ECFCs under normoxic or hypoxic conditions. $n = 4$–5 biological donors, each assayed in a separate chip. Bars are means ± SEM; one-way ANOVA with a Tukey post-test; *$p \leq 0.05$. Volcano plots show **e** DEG in PAH ECFCs with BMPR2 mutations vs. healthy controls in normoxia, **f** DEG in PAH ECFCs vs. healthy controls in hypoxia (ECFC PAH model), **g** DEG in HPASMCs in hypoxia. In **e**–**g** $n = 5$ biological donors, each in a separate chip.

key regulators of vascular homoeostasis such as *DDAH1, CAV1, PDGFB, KLF2, APLN* (Supplementary Data 6). HPASMCS co-cultured with PAH ECFCs showed 972 DEG (Supplementary Fig. 16) enriched in cell cycle progression (E2F targets), G2M checkpoint, oxidative phosphorylation, interferon-gamma pathways (Hallmark, FDR < 0.01). A list of DEG in BMPR2-deficient ECFCs and HPASMCs co-cultured with these cells under normoxic conditions is provided in Supplementary Data 7.

PAH ECFCs from the two-hit ECFC PAH model (combining PAH *BMPR2* mutations and hypoxia) showed 2360 DEG, while HPASMCs showed 689 DEG (logFC > 0.25, $P < 0.05$) (Fig. 4f, g and Supplementary Data 1). Hierarchical gene clustering in

ECFCs and HPASMCs under different experimental conditions revealed genotype- and oxygenation-specific changes in gene expression (Fig. 5a, b and Supplementary Fig. 17).

ECFC DEG from the two-hit model showed enrichment in key PAH pathways, including angiogenesis, apoptosis, EMT, fibrosis, inflammation, proliferation, nitric oxide, TGF-β, Notch and Wnt signalling (Hallmark, FDR < 0.01) (Fig. 5a and Supplementary Fig. 18a). Only ~30% of these genes were related to *BMPR2* mutation. The remaining ~70% of genes induced by additional hypoxic exposure showed links with glycolysis, adipogenesis, cell cycle, oxidative phosphorylation, Rho signalling, TNF-α signalling and cellular senescence (Supplementary Data 8).

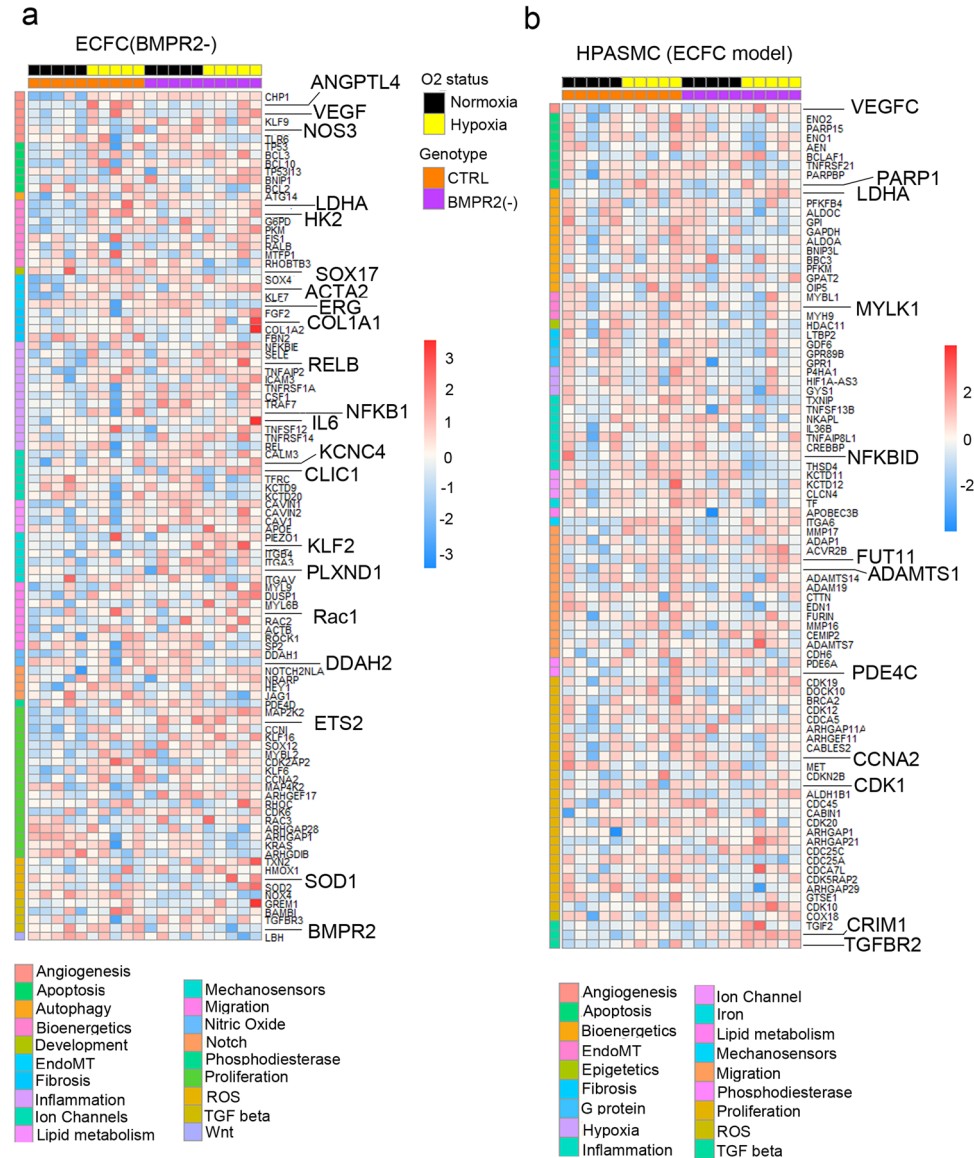

**Fig. 5 Disease pathways in control and patient-derived ECFC datasets.** Top DEG associated with cardiovascular diseases were grouped into categories and visualised as TPM heatmaps. Colour-coded pathways are shown underneath the heatmaps, with key gene symbols enlarged in **a** PAH ECFCs and **b** HPASMCs co-cultured with ECFCs. The genotype status (control and BMPR2 knockdown) and oxygen status (normoxia or hypoxia) are shown at the top of the heatmap in different colours, as indicated. Changes in gene expression are also colour coded, with blue denoting a lower relative gene expression and red denoting a higher relative gene expression. Each column represents 1 donor; $n = 5$ different biological donors/treatment groups.

HPASMC genes from the two-hit ECFC PAH model showed significant associations with angiogenesis, apoptosis, hypoxia, inflammation, proliferation, fibrosis and TGF-β pathways (Fig. 5b and Supplementary Fig. 18b). 38% of these genes could be related to the BMPR2 loss of function, with 62% of changes affected by hypoxia and linked with glycolysis, pentose metabolism, cell cycle, and immune responses. A list of HPASMC genes affected by ECFC *BMPR2* mutations is provided in Supplementary Data 6. Hypoxia-regulated genes in these cells and corresponding GSEA pathway analysis data are shown in Supplementary Data 2 and 9, respectively.

**Transcriptomic overlaps between different PAH datasets.** RNAseq datasets from the two microfluidic models of endothelial dysfunction in PAH were compared with published gene datasets from IPAH lung transplants[22,23] and other gene databases with known PAH associations[24]. DEG from our models and other published datasets are listed in Supplementary Data 10 and 11. The results of the comparative analysis are shown in Fig. 6.

The highest number of shared endothelial genes was found between our two microfluidic PAH models (212 genes), between the microfluidic ECFC PAH model and the IPAH PAEC gene dataset (63 genes) and between the two-hit PAH model and the IPAH PAEC dataset (33 genes)[24] (Fig. 6a and Supplementary Data 11). GO pathway analysis (Hallmark) of the overlapping endothelial gene datasets revealed their associations with key PAH pathways, including EMT, TNF-α signalling via NFkB, hypoxia, apoptosis, and p53 pathway (FDR < 0.05) (Supplementary Data 12).

Consistently, the highest number of shared differentially expressed SMC genes was found between our microfluidic PAH models (94 genes), between the two-hit ECFC PAH model and IPAH PASMCs[25] (86 genes) and between the two-hit HPAEC

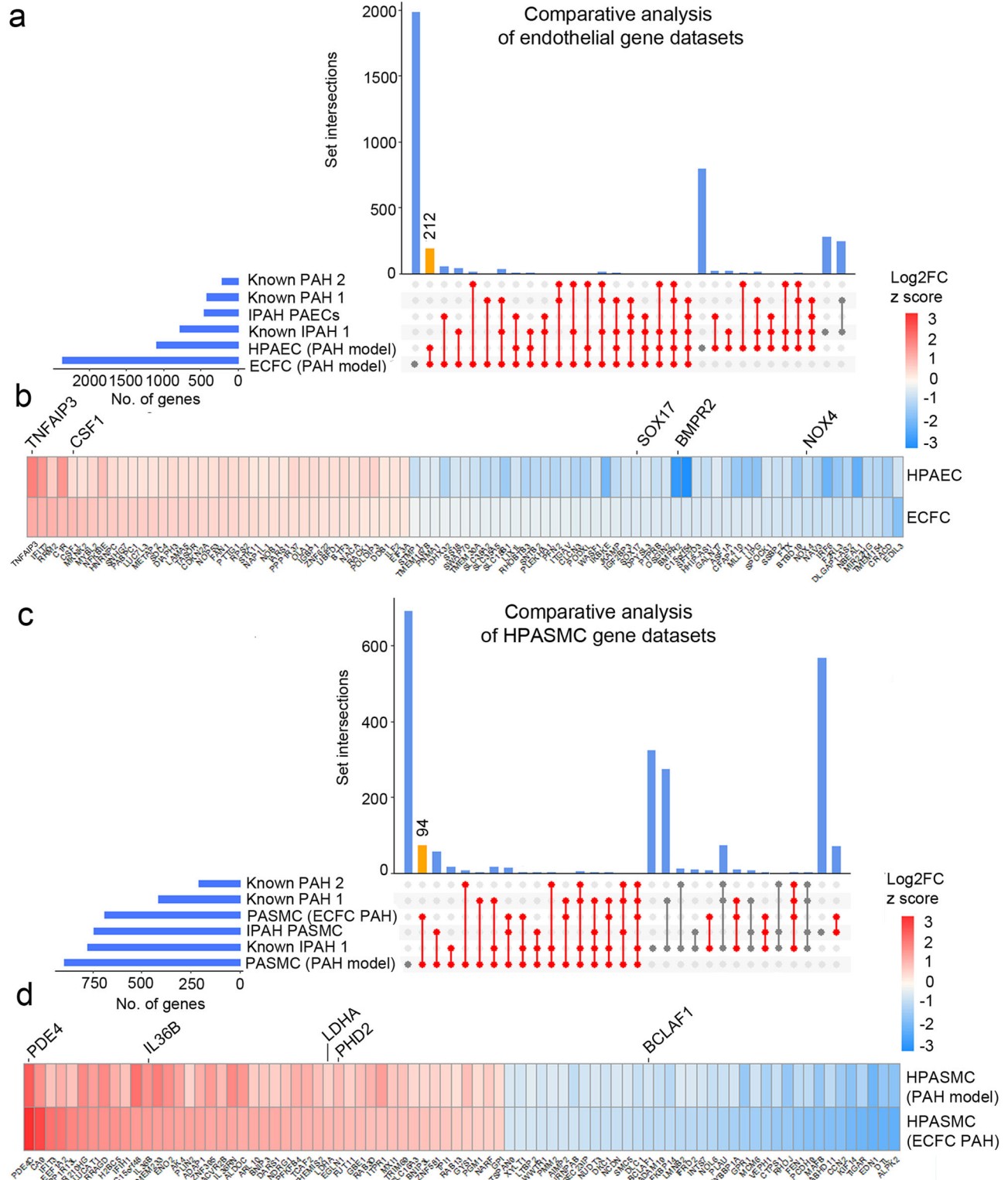

**Fig. 6 Comparative analysis of microfluidic PAH models and published PAH and IPAH gene datasets.** DEG datasets from the adenoviral two-hit models (PAH Model) and patient ECFC models (ECFC PAH) of PAH were compared against previously reported RNAseq studies and lists of genes known to be associated with PAH and IPAH. UpSet plots and heatmaps detail comparative analysis of **a**, **b** endothelial gene datasets and **c**, **d** smooth muscle cell datasets. In **a**, **c** the largest gene overlaps are highlighted in yellow. Heatmaps in **b**, **d** visualise the similarity of gene expression changes between the two microfluidic models of PAH, the two-hit PAH model and the ECFC PAH model; downregulated genes are in blue and upregulated genes are in red. RNAseq datasets used for comparative analysis were from PASMCs from IPAH lung transplants[22] (database named here IPAH PASMCs); PAECs isolated from IPAH lung transplants[23] (named here IPAH PAEC). Other comparative gene sets included genes with known PAH-associations [24] (here named "known PAH2") and the DisGENET public database https://www.disgenet.org/search, where following gene sets were obtained: DisGENET PAH (https://www.disgenet.org/browser/0/1/0/C2973725/), (here named "known PAH1"); DisGENET IPAH (https://www.disgenet.org/browser/0/1/0/C3203102/) (here named "known IPAH1").

PAH model and IPAH PASMCs (76 genes) (Fig. 6c and Supplementary Data 10). The shared SMC gene set showed links with TGF-β signalling, glycolysis, fatty acid metabolism, pentose phosphate pathway, TNF-α via NFκB signalling, cell cycle/G2M checkpoint (GO pathway analysis, FDR < 0.05) (Supplementary Data 13).

**Endothelial SOX17 regulates PASMC proliferation**. Loss of endothelial SOX17 was one of the most prominent changes seen in BMPR2-deficient cells in microfluidic models of PAH and PAH databases. Consistently, expression of SOX17 was markedly reduced in the lung endothelium of mice carrying a disabling ligand-binding domain mutation of *BMPR2* (BMPR2^C118W)[26] (Fig. 7a–c and Supplementary Fig. 19). Reduced nuclear localisation of SOX17 was also noted in hypoxic, BMPR2-deficient HPAECs (Fig. 7d and Supplementary Fig. 20).

To investigate the potential role of endothelial SOX17 in the regulation of PASMC proliferation, a compensatory overexpression of SOX17 was induced in BMPR2-deficient HPAECs. SOX17 prevented an increase in PASMCs proliferation under the two-hit conditions (Fig. 7e). Quantitative proteomic analysis of SOX17-overexpressing HPAECs identified 20 potential mediators of this response, including prostacyclin synthase (*PTGIS*), prostaglandin reductase (*PTGR2*), superoxide dismutase 1 (*SOD1*), Dickkopf WNT Signalling Pathway Inhibitor 2 (*DKK2*) and Krüppel-like factor 16 (*KLF16*). An increase in prostacyclin synthase expression was the most significant change noted (2,5-fold increase, *P* = 0.006) (Fig. 7f). Normalised and raw protein abundances in study groups are provided in Supplementary Data 14–16.

Selective prostacyclin receptor inhibitor RO1138452[27] abolished the anti-proliferative actions of SOX17, reaffirming the key role of prostacyclin in the regulation of PASMC proliferation under the two-hit disease conditions.

Prostacyclin exerts vasodilatory and anti-proliferative effects on vascular SMCs and its deficiency has been linked to the reduced expression of *PTGIS* in PAH[28,29]. We identified eight predicted SOX17-binding sites within the *PTGIS* promoter and eleven binding sites within its putative enhancer regions, suggestive of direct regulatory interaction (Supplementary Fig. 21). Multiple SOX17-binding sites were also found in promoters and enhancers of other PAH-relevant genes, including *PTGIS*, *PTGR2*, *DKK2*, except for *SOD1*, which included only one predicted SOX17-binding site in one of its enhancers.

**Drug testing in microfluidic two-hit models of PAH**. To test if PASMC proliferation induced under the two-hit conditions is responsive to treatment with PAH-relevant drugs, Ambrisentan, an FDA-approved selective antagonist of ET-1 receptor A and AZD5153, a compound from a structural class of BET inhibitors, were added to the flow system. Two BRD-binding motifs (warheads) of AZD5153[29] enable bivalent-binding and confer high potency for BRD4, differentiating this drug from alternatives including Apabetalone (RVX-208)[12]. At clinically relevant[10,12] concentrations both Ambrisentan and AZD5153 effectively inhibited PASMC proliferation (Fig. 7g, h).

## Discussion

We present an organ-on-chip model of vascular endothelial and smooth muscle cell interactions that permits the study of dynamic changes in molecular and functional cell phenotype under physiological flow conditions in response to key factors linked to the development of PAH, such as *BMPR2* silencing and hypoxia. The study describes pulmonary endothelial cell phenotype changes required for the induction of medial smooth muscle activation and proliferation and highlights the potential importance of a link between BMPR2 and an arterial identity determinant, SOX17 in the regulation of this response. The model can accommodate blood-derived endothelial cells from patients, offering the prospect of tailoring medicines to individual patients. PASMC proliferation, a key feature of medial hyperplasia, was attenuated by five drugs exhibiting different modes of action, validating the use of this platform for drug testing. The experimental design and key findings are summarised in Fig. 8.

Most *BMPR2* mutation carriers and heterozygous *BMPR2* knockout animals do not spontaneously develop PAH and a second hit is required for the full manifestation of the disease[4]. Consistent with this premise, a combination of endothelial *BMPR2* knockdown with hypoxia in PA-on-a-chip was required to induce HPASMC proliferation. The mechanisms by which endothelial BMPR2 depletion primes vascular cells for the disease and converges with the effects of the second hit are not well understood. Using a combination of functional assays, RNA sequencing and pathway analysis in human pulmonary endothelial and smooth muscle cells and cells from PAH patients with disabling *BMPR2* mutations under the two-hit conditions, we have identified a microfluidic signature of vascular responses characteristic of PAH. Crucially, this gene dataset showed a substantial overlap with transcriptomic changes reported in PAH and identified gene targets of potential interest.

Loss of BMPR2 in HPAECs inhibited expression of TGF-β family genes involved in vascular differentiation, angiogenesis and vessel maturation[30], consistent with changes reported in PAH. HPAECs also showed reduced expression of junctional markers *CDH5* (VE-cadherin), *GJA5* (connexin 40) and several integrins, likely to predispose the endothelium to stress-induced damage.

BMPR2-deficient HPAECs also showed reduced expression of NO bioavailability enzymes, *NOS3, DDAH1* and *DDAH2*, consistent with PAH vasculopathy[31,32]. Other key PAH-related changes included a reduction in arterial identity factors *SOX17, SOX18*, and *KLF2* as well as regulators of angiogenesis, proliferation, and endothelial sprouting behaviour. Exposure of these BMPR2-deficient endothelial cells to hypoxia altered the expression of genes controlling oxygen transport, cell cycle, transcriptional regulation and inflammation, including genes with well-documented links to PAH: *KCNK*[33,34], *HDAC4*[35], *BRD4*[12], *CAMK2G*[36,37], and *ICAM1*[38]. To summarise, the two-hit microfluidic model offers insight into processes seen during the development of the disease, associated with a loss of arterial identity, reduced eNOS signalling and increased susceptibility to damage, accompanied by inflammation and metabolic shift towards glycolysis.

Differential gene expression in HPASMC co-cultured with BMPR2-deficient HPAECs in the two-hit PAH model was enriched for disease-specific phenotypes associated with angiogenesis, apoptosis, inflammation, vasoconstriction and TGF-β signalling. Downregulation of extracellular matrix genes (collagens, fibronectin, laminin, tenascin), modulators of SMC migration and proliferation (serpin, periostin, caveolin[39] and apelin[40]) and upregulation of matrix metalloproteinases is likely to compromise arterial structural integrity and stimulate SMC migration and proliferation[41,42]. Another notable observation involved a reduction in plasmalemmal potassium channels, *KCNMA, KCNAB2*, and *KCNK1* in these cells, a change linked with pulmonary vasoconstriction and vascular remodelling[43]. Consistent with the pro-proliferative phenotype of HPASMCs under the double hit conditions, we observed a 6-fold increase in the activity of hexokinase, a key enzyme that controls the rate of glycolysis and pentose cycle, the two key pathways which provide energy and metabolites for the rapidly growing cells.

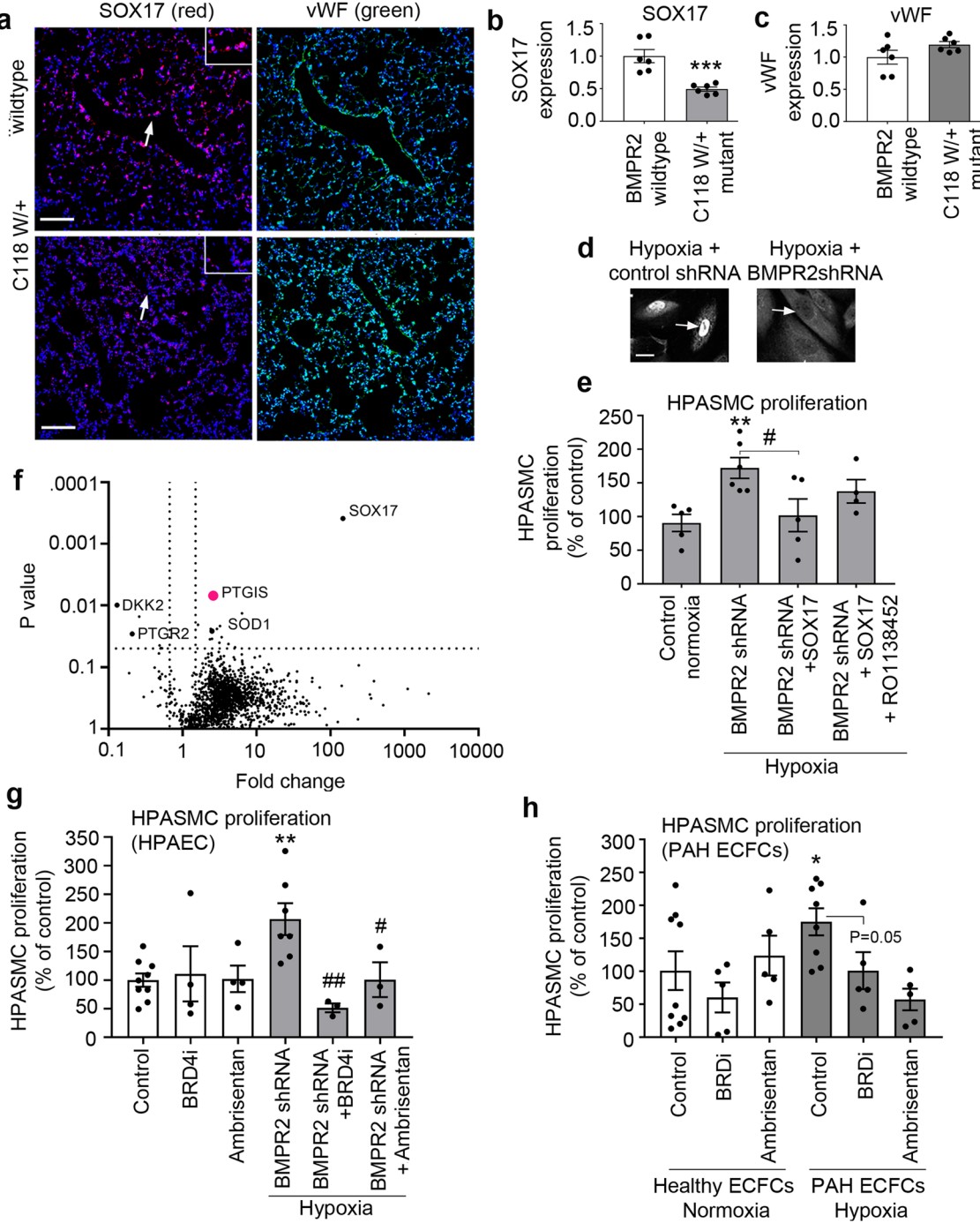

**Fig. 7 Endothelial SOX17 regulates PASMC proliferation in the two-hit PAH model. a** Representative confocal images of lung sections of wildtype and BMPR2 (C118W/+) mice with SOX17 (red) and vWF marking endothelium (green) and nuclei in blue (DAPI), as indicated; Bar = 50 μm. Arrows point to cell nuclei. Magnified images of boxed areas are shown in the top right corner. **b**, **c** Corresponding graphs showing SOX17 and vWF expression in mouse lung tissues, as indicated. **d** Confocal images showing localisation of SOX17 in BMPR2-deficient HPAECs (BMPR2 shRNA) and controls (control shRNA), as indicated. Arrows point to cell nuclei. Bar = 10 μm. **e** Volcano plot showing SOX17-induced DE proteins, with prostacyclin synthase (PTGIS) marked in pink and selected proteins of interest highlighted. **f** Effect of endothelial SOX17 overexpression on HPASMC proliferation in the two-hit model (BMPR2shRNA + hypoxia) model, with and without prostacyclin receptor inhibitor, RO1138452 (10 μM). **g**, **h** Effect of Ambrisentan and BRD4 inhibitor AZD5153 on HPASMC proliferation in two-hit models of PAH, utilising BMPR2-deficient HPAECs or PAH ECFCs, as indicated. In **b**, **c** n = 6/group. ***P < 0.0001, Student t-test; In **e** n = 4–6, *P < 0.05 comparison with normoxic control, #P < 0.05, comparison as indicated; one-way ANOVA with Tukey post-test. In **g** n = 4–8, in **h** n = 5–10.

Patient blood-derived endothelial colony-forming cells (ECFCs) display abnormalities characteristic of PAH, raising interest in their application in personalised medicine[20,21]. To validate our model, we used patient ECFCs with disabling *BMPR2* mutations in place of BMPR2-depleted HPAECs. The comparative analysis of cell responses revealed several important similarities between the two cell types. Consistent with changes seen in BMPR2-deficient HPAECs, ECFCs with *BMPR2* mutations showed upregulation of

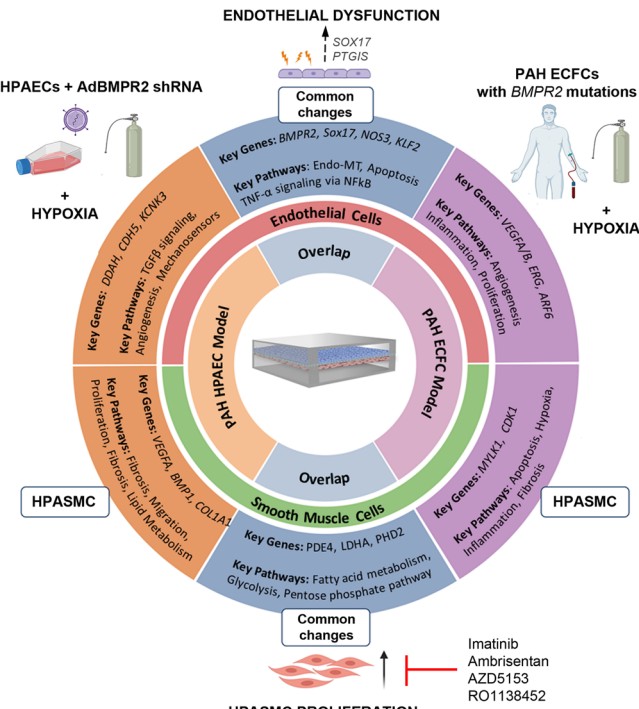

**Fig. 8 Microfluidic two-hit model of PAH.** AdBMPR2 shRNA-treated HPAECs and PAH ECFCs with disabling *BMPR2* mutations were co-cultured with HPASMCs under hypoxic (2% O$_2$) conditions in PA-on-a-chip for 24 h. The diagram shows selected key PAH genes and pathways identified by transcriptomic profiling of endothelial and smooth muscle cells.

inflammatory genes (*IL6, REL, NFKB1, NFKB2, CXCL1, 2, 3, 8, TRAF1, ICAM3*), glycolysis (*HK2, LDHA, PDK4*) and mitochondrial function (*FIS1, STOML2*)[44] and downregulation of key markers of arterial identity (*ERG* and *SOX17*)[45]. However, in contrast with the response seen in HPAECs, ECFCs showed an upregulation of genes regulating NO bioavailability (*NOS3, DDAH1, DDAH2*) and increased expression of genes promoting endothelial repair and angiogenesis (*VEGFA, VEGFB, FGF2, CLIC1, RAC1, ROCK1, ARF6, KLF2, KLF6, KLF7*), which may be reflective of the uncontrolled propagation of reparative endothelial cell responses in advanced PAH[46].

A link between *BMPR2* and *SOX17* reported in this study, is intriguing, considering the association of variation in both genes with susceptibility to PAH[5,47]. The loss of endothelial SOX17 was necessary for the induction of PASMCs proliferation under the two-hit conditions and we identified prostacyclin synthase (*PTGIS*) and prostacyclin, as likely mediators of this response. Establishing the precise nature of *SOX17* interactions with *PTGIS* and other genes involved in pathological processes in PAH such as *PTGR2*[48], *SOD1*[49], Wnt pathway modulator, *DKK2*[50,51] or lipid metabolism and insulin resistance regulator, *KLF16*[52] is beyond the scope of this study and will require further investigation.

The clinically approved endothelin receptor antagonist, Ambrisentan, and a novel drug candidate, BRD4 inhibitor, AZD5153 were effective in inhibiting PASMC proliferation in the inducible microfluidic models featuring BMPR2-deficient HPAECs and patient-derived ECFCs. BRD4, a member of the BET (bromodomain and extra-terminal motif) family, is a key transcriptional modulator of cell apoptosis and proliferation and a critical epigenetic driver for cardiovascular diseases[12]. Consistent with the observations in PAH lungs[12], expression of BRD4 was significantly elevated in HPAECs cultured under the two-hit conditions, compared with healthy, untreated controls.

The economic use of cells, short experimental time, device scalability, and low production and operation cost make our PA-on-a-chip an attractive model for drug testing. The only other existing PAH-on-a-chip model[53,54] included intimal, medial, and adventitial cells from PAH pulmonary arteries grown in straight channels arranged side-by-side, with cell-to-cell contact restricted to the gaps between silicone posts situated between the channels[54]. This model can provide valuable information regarding paracrine regulation of cell migration but cannot evaluate endothelial barrier function and is limited by poor accessibility of patient lung material, which precludes its wider application in research and drug testing. Our device, which accommodates only two cell types, is unlikely to fully reflect the complex pathophysiology of PAH and therefore future sourcing of patient-derived cells and introducing other cell types will be a vital consideration. The current design should be further improved to allow the migration of cells between the two vascular layers, a response linked to vascular remodelling in PAH. One such approach would be increasing the size of pores in the membrane separating the two microfluidic channels.

Animal models have been instrumental in deciphering the molecular mechanisms underlying the disease but are often criticised for not fully reproducing the pathology of human disease and for low success in the clinical translation of new drugs[8]. The benefits and limitations of animal models of PAH as well as new measures undertaken to improve their translational value have been summarised in the two excellent recent reviews[8,55,56]. Our model successfully reproduced the disease characteristics previously described in human and animal PAH, such as medial cell proliferation and activation of PAH-associated pathways in pulmonary arterial intimal and medial cells. However, considering the limitations of this model, such as lack of complexity, variability associated with genetic diversity, age and comorbidities of donors[56], as well as potential differences in manufacturing protocols amongst laboratories, attention to stringent study design[56], including data validation with robust animal models, may be required in future device applications in exploratory and confirmatory PAH preclinical studies.

In summary, we provide a microfluidic and inducible model of vascular cell dysfunction, informing of functional and transcriptomic effects of endothelial BMPR2 deficiency and hypoxia. Significant overlaps with pre-existing lists of differentially expressed PAH genes, are amenable for target discovery/validation and can potentially be applied for use in drug screening or toxicology.

## Methods

**Photolithography, soft lithography and chip fabrication.** Individual photomasks were designed for endothelial and smooth muscle cell chambers using AutoCAD 2017 (Autodesk, CA, USA) and printed on high-resolution photomask films (Micro Lithography Services Ltd, UK). Photolithography, soft lithography and pulmonary artery-on-a-chip fabrication were conducted in an ISO Class 5 Cleanroom (<100,000 particles/m$^3$). A detailed description of fabrication steps is provided in Supplementary Methods.

**PA-on-a-chip simulation.** Finite element method-based simulations were used to simulate the fluid–structure interaction within the PA on-a-chip device using COMSOL Multiphysics version 4.4. A detailed description of the model developed is provided in Supplementary Methods. PDMS walls model parameters are shown in Supplementary Table 1.

**Cell culture.** HPAECs (PromoCell, Cat. no. C-12241) from 3 to 4 different biological donors were cultured in T75 tissue culture flasks (Sarstedt, Germany, Cat. no. 833911002) coated with 10 µg/mL Fibronectin (10 µg/mL, EMD Millipore Corp, USA; 341631) in endothelial cell growth medium 2 (ECGM-2, Promocell, Germany C-22211), supplemented with 2% foetal calf serum (FCS) and growth factor mix (PromoCell, Cat. no. C-22111), 1% Streptomycin/Penicillin (100 µg/mL) and 1% MycoZap™.

HPASMCs (Lonza, Cat. No. CC-2581) were cultured in T75 tissue culture flasks coated with 0.2% porcine gelatin (Sigma, Cat. no. G1890) in smooth muscle cell growth medium 2 (SmGM-2, PromoCell, Cat. no. C-22062), supplemented with 5% FBS and growth factor supplement (PromoCell, Cat. no. C-39267) and antibiotics. Cells were acquired at passage 3 and were used for experiments in passages 4–6. Donor information is provided in Supplementary Table 2. For HPAEC and HPASMC co-culture, ECGM-2 medium was supplemented with 10% foetal bovine serum (FBS). VEGF, ascorbic acid, heparin, and hydrocortisone, known to inhibit HPASMC proliferation, were excluded.

To induce hypoxia, cells were incubated in a humidified 37 °C incubator set at 5% $CO_2$ and 2% $O_2$.

**Blood-derived human endothelial cells.** Human endothelial colony forming cells (ECFCs) were derived from peripheral blood samples and characterised, as previously described[57]. Venous blood samples were obtained with local ethics committee approval and informed written consent (REC Ref. 17/LO/0563) from healthy volunteers ($n = 5$) and HPAH patients with rare pathogenic BMPR2 variants ($n = 5$). ECFCs were cultured in 1% porcine gelatin-coated T75 tissue culture flasks (Sigma, Cat. no. G1890) in EGM-2 medium (CC-3156, Lonza Biologics, Slough, UK), supplemented with growth factors (CC-4176, EGMTM-2 bullet kit, Lonza), 10% FBS (HyClone, Thermo Scientific, South Logan, UT, USA) and 1% antibiotic/antimycotic solution (Gibco, Invitrogen, Paisley, UK). All ECFCs were used between passages 3 and 6.

Demographic and clinical features of healthy subjects and PAH patients are shown in Supplementary Table 3 and IPAH BMPR2 mutation variants are listed in Supplementary Table 4.

**HPAEC and HPASMC co-culture in Transwell filters.** HPAEC and HPASMC were co-cultured in Transwell insert containing a polyethylene terephthalate (PET) membrane with 0.4 μm pores and a pore density of $(2.0 \pm 0.4) \times 10^6$ pores/cm² (Cat. no. 353180, Corning, USA). A detailed protocol is provided in Supplemental Methods.

**On-chip cell co-culture.** Microfluidic channels were sterilised during manufacture using oxygen plasma and additionally sterilised under UV light for 30 min. Inlets/outlets of channels not used in this procedure were covered with 3 M Magic Tape. Chips selected for use were flushed with 100% ethanol, followed by two washes with sterile PBS. Before seeding cells, microfluidic channels were filled with sterile 100 μg/mL rat tail collagen (BD Biosciences, UK; Cat. no. 354236) in 20 mM acetic acid (Honeywell, UK; Cat. no. 695092) and incubated for 1 h at room temperature. Following coating, 150,000 HPASMCs suspended in 15 μL of co-culture medium were seeded into the smooth muscle (bottom) channel. The chip was turned over and placed in a humidified 37 °C incubator for 2 h to allow cell attachment. The smooth muscle chamber was then gently flushed with a co-culture medium to remove unattached cells and the chip was reverted to the upright position. 150,000 HPAEC or ECFCs suspended in 15 μL (10,000 cells/μL) of co-culture medium were seeded into the endothelial (top) channel and incubated for 2 h at 37 °C. The endothelial cell and smooth muscle cell channels were then gently flushed with 200 μL co-culture media to remove non-adherent cells, before connecting access ports to custom media reservoirs (Cambridge Glassblowing, UK) via 0.8 mm ID flexible tubing (IBIDI, Germany, cat. no. 10841). The medium was perfused using a REGLO ICC 4 channel 12 Roller Pump (Cole-Parmer, UK, cat. no. ISM4412), utilising Ismatec Pump Tubing, PharMed® BPT, 0.76 mm ID; 100 ft (Cole-Parmer, UK, cat. no. WZ-95809-24) and Ismatec Pump Tubing, 3-Stop, PharMed® BPT, 0.76 mm ID (Cole-Parmer, UK, cat. no. WZ-95714-24). 0.8 mm ID Y-Style polypropylene barbed flow splitters were used to partition flow between two chips in an independent flow circuit (IBIDI, Germany, cat. no. 10827). All chips were perfused with 6 dynes/cm² flow.

**Manipulation of BMPR2 and SOX17 expression in HPAECs.** ~90% confluent HPAECs grown in six-well plates were infected with Ad-Control-shRNA-GFP (Vector Biolabs, USA; Cat. No. 1122) or Ad-BMPR2-shRNA-GFP (Vector Biolabs, USA; Cat. No. shADV-202207) (titre $1 \times 10^8$ PFU/mL) at the multiplicity of infection (MOI) of 1:750. Three hours later, media were changed and cells were incubated for 24–48 h before further experimentation. Transfection efficiencies were determined by fluorescent cell counting and BMPR2 knockdown was confirmed by qPCR and western blotting. In some experiments, AdSOX17 (Vector Biolabs, USA, Cat no. ADV-224019) was added for 24 h at the MOI 1:500 to induce compensatory overexpression of SOX17.

**RNA isolation, reverse transcription and quantitative polymerase chain reaction (qPCR).** Detailed experimental procedures are described in Supplemental Methods. Briefly, cells were washed in PBS, trypsinised and spun down for 5 min at 5000×g. Total RNA was extracted using either a Monarch Total RNA Miniprep Kit (Cat. no. T2010S; New England Biolabs, MA, USA) or RNeasy Plus Micro Kit (Cat. no. 74034; Qiagen, Germany). Double elutions were performed to maximise RNA recovery from spin columns. Total RNA levels were quantified using a NanoDrop™ ND2000 spectrophotometer (Thermo Scientific, UK).

50 ng (from chips) or 100 ng (from all other samples) of extracted RNA was reverse transcribed into cDNA using a LunaScript RT SuperMix Kit (Cat. no. E3010L; New England Biolabs, MA, USA). Components of the RT reaction mix and RT cycle are detailed in Supplementary Tables 5 and 6. qPCR master mix components and qPCR cycle conditions are listed in Supplementary Table 7.

All primer sequences were designed from FASTA sequences (PubMed, NCBI). The list of primers is shown in Supplementary Table 8.

**RNA sequencing.** Next-generation RNA sequencing of HPAECs transfected with Ad-shCTL or Ad-shBMPR2, HPASMCs, healthy volunteer ECFCs and PAH ECFCs with 4–5 biological replicates, was performed at the Imperial British Research Council Genomics Facility (Imperial College London, UK). RNA quality and quantity were determined using a Tapestation System (Agilent Technologies, UK). 50 ng of high-quality total RNA (RNA Integrity Number Score ≥ 8.0) was used as starting material. Polyadenylated mRNA enrichment was performed using a NEBNext® Poly(A) mRNA Magnetic Isolation Module (Cat. no. E7490L; New England Biolabs, MA, USA). RNA libraries were prepared using a NEBNext® Ultra™ II Directional RNA Library Prep Kit (Cat. no. E7760L; New England Biolabs, MA, USA), in accordance with the manufacturer's protocol. Samples were indexed at the PCR enrichment stage of RNA library preparation with NEBNext Multiplex Oligos for Illumina 96 Unique Dual Index Primer Pairs (Cat. no. E6440L; New England Biolabs, MA, USA). RNA sequencing was then performed on an Illumina Hiseq4000 Paired End 75 with dual indexing (Illumina, CA, USA) yielding ~25–30 million reads per sample.

Raw sequencing data (fastq files) were aligned to a reference human transcriptome (GENCODE release 34) using Salmon (v1.4.0)[58] to produce transcript abundance estimates. Estimates were converted to gene expression data in RStudio (https://www.R-project.org/) using the R package tximport. Differential gene expression was performed using edgeR (v3.30.3)[59], which utilises the limma R package (v3.44.3)[60] adapted for optimising performance in selecting differentially expressed genes in relatively low numbers of biological replicates. This analysis was corrected for at least two principal components, which each component individually explaining >1% of the variance in the dataset. Genes were considered differentially expressed if the P value was <0.05 and the absolute fold change was >0.25. Functional annotation and enrichment of genes were performed using GSEA software (Broad Institute, MIT; UC San Diego) and Ingenuity Pathway Analysis (IPA, Qiagen) using in-built false discovery rate multiple test corrections. Prior to GSEA analysis, genes that met DGE thresholds were pre-ranked following the guidelines from Reimand et al.[61] and were analysed using the following gene set collections from the Molecular Signatures Database (MSigDB): Hallmark[62], KEGG[63], or Reactome[64]. The following R packages were used to produce selected graphs: Volcano plots using EnhancedVolcano (https://github.com/kevinblighe/EnhancedVolcano); dot plots using ggplot2 (https://ggplot2.tidyverse.org); heatmaps using pheatmap (https://cran.r-project.org/package=pheatmap).

**Comparative analysis with published RNAseq datasets.** RNAseq datasets used for comparative analysis were from HPASMCs isolated from IPAH lung transplants[22] and HPAECs isolated from IPAH lung transplants[23]. Other databases of genes with known PAH associations were downloaded from Supplementary Data 4[24] (Disgenet public website https://www.disgenet.org/search; DisGENET PAH (https://www.disgenet.org/browser/0/1/0/C2973725/), DisGENET IPAH (https://www.disgenet.org/browser/0/1/0/C3203102/.

**Proteomic analysis.** Protein concentration of HPAECs samples ($n = 3$/group) was determined by Bradford assay and protein was digested for MS analysis by single-pot digestion. Digestion was halted by the addition of TFA and immediate desalting. Desalting was performed using HLB uElution plates (Waters) with a vacuum manifold. Desalted peptides were eluted, dried, and quantified in peptide assay. Subsequently, samples were adjusted to 0.1 μg/μl for LC–MS and injected into a 180 μm × 20 mm C18 trap column (NanoEase M/Z Symmetry, Waters). Peptides were separated on a 75 μm × 200 mm C18 column (nanoEase M/Z HSS T3, Waters) on an Acquity M-Class UPLC (Waters) interfaced with a Synapt G2S mass spectrometer (Waters). Data were acquired in positive resolution mode, using HDMSE over 50–2000 m/z with a scan time of 0.5 s with lock mass acquired for 3 scans every 1 min (Leucine-enkephalin). For high-energy scans, the collision energy was ramped between 19 and 45 V with a cone voltage of 40 V. Samples were randomised and each sample was interspaced with a blank to prevent carryover.

Data were processed using Progenesis QI for Proteomics with lock mass correction (556.2771 Da), with peak picking performed with default settings, but maximum charge set to 6. Data were searched against a human reference proteome (Swissprot, downloaded on 28/06/2021) with trypsin digestion and up to 2 missed cleavages specified. Methionine oxidation was set to variable and cysteine carbamidomethylation was set as a fixed modification. FDR filtering was performed to 4%, with 3 fragments per peptide, 5 fragments per protein, and 1 peptide per protein.

**Identification of SOX17 binding sites.** The promoter region coordinates of the genes investigated were chosen based on the GeneHancer Identifiers[65] provided in Genecards[66]. A detailed description of the analysis is provided in the Supplemental Methods.

Predictions were made using human as well as mouse SOX17 motifs, which share identical DNA-binding domains, with the use of the human gene CIS-BP library. SOX17-binding sites in promoter and enhancer regions of the investigated genes are shown in Supplementary Tables 9 and 10.

**Permeability assays**. Endothelial permeability in Transwell dishes and in PA-on-a-chip was evaluated spectrophotometrically by measuring the passage of 1 mg/mL 40 kDa FITC-Dextran (FD40S, Sigma Aldrich, Dorset, UK) across the endothelial and smooth muscle layer[57]. In some experiments, 1 U/mL thrombin (T7513, Sigma Aldrich, Dorset, UK) was added to the co-culture media. A detailed description of experimental protocols is provided in Supplemental Methods.

**Immunostaining**. Cell fixation and immunostaining protocols are described in Supplemental Methods. Histological sections of lung tissues from treatment-naïve PAH patients at lung transplantation ($n = 2$), and control tissues comprising uninvolved regions of lobectomy specimens from 2 unused donor lungs were from the tissue archives at Hammersmith Hospital, Imperial College London. Detailed immunostaining procedures and antibody lists are provided in Supplemental Methods. A list of primary and secondary antibodies used is shown in Supplementary Table 11.

**BMPR2C118W Knockin Mouse**. This study used lung sections from published study[26]. All work with animals was conducted in accordance with the UK Animals (Scientific Procedures) Act 1986 and approved under Home Office Project License 80/2460.

**EdU proliferation assay**. Cell proliferation was quantified using an EdU Cell Proliferation Assay Kit (EdU-594, EMD Millipore Corp, USA, Cat. No. 17-10527) according to the manufacturer's instructions. EdU-positive cells were visualised under a fluorescent Zeiss AxioObserver widefield microscope (Carl Zeiss AG, Germany). Images and z-stacks were taken with ×10 objective and were analysed with ImageJ software. Data are presented as a percentage of the number of EdU-positive cells vs. the total cell number.

**Hexokinase activity measurement**. Hexokinase activity was measured in pooled samples of PASMCs ($1.5 \times 10^5$ cells/treatment) using Hexokinase Assay Kit (Abcam, ab136957), according to the manufacturer's protocol.

**Drug treatment**. Imatinib Mesylate (Enzo life sciences, UK; Cat. no. ALX-270-492-M025) and Ambrisentan (Astra Zeneca) were added directly to the cell culture medium at clinically relevant concentrations of 10 μM[67] and 1.25 nM[10], respectively. BRD4 inhibitor, AZD5153 (Astra Zeneca) was added at the concentration of 16.5 nM, based on the pre-clinical use of BET inhibitors[12] and the in vitro dose-response experiments. Cells were incubated with the inhibitors for 24 h under normoxic (21% $O_2$) or hypoxic (2% $O_2$) conditions accordingly.

**Apoptosis assay**. Apoptosis was measured using a Click-iT Plus TUNEL Assay for In Situ Apoptosis Detection Kit with Alexa Fluor 647 dye (Life Technologies, Cat. no. C10619), as per the manufacturer's instructions. TUNEL-positive cells were visualised under a fluorescent Zeiss AxioObserver widefield microscope (Carl Zeiss AG, Germany). Images and z-stacks were taken under ×10 objective and analysed with ImageJ software. Data are presented as a percentage of the number of TUNEL-positive cells vs. total cell number.

**Cell counting**. 3D Z-stacks of tiled, 5 μm-thick image slices of microfluidic channels were acquired under AxioObserver widefield microscope. In microfluidic channels, endothelial cells (HPAECs or ECFCs) were labelled green: the cells were either overexpressing AdGFP or their nuclei were immunostained with anti-Erg antibody. Nuclei of proliferating cells (HPAEC/ECFCs and HPASMC) were labelled red (EdU-594) and nuclei of all cells were labelled blue (DAPI). Image stacks were flattened and three 500 μm × 1000 μm regions of interest were selected at the front, middle and at end of each channel, within the area of HPAEC/HPASMC overlap. To facilitate cell counting, a custom ImageJ script was developed that generated a binary black/white mask of fluorescent signatures (cells and nuclei) within each colour channel. Green cells with blue nuclei were classed as quiescent ECs, green cells with blue/red nuclei were classed as proliferating ECs and the remaining cells (not green) with blue nuclei only and blue/red nuclei only, were classed as quiescent SMCs and proliferating SMCs, respectively. Counts were made within each colour channel before overlaying images with DAPI binary mask to produce final counts. Selected counts were additionally checked by manual scoring of cells in each fluorescent channel using the Grid and CellCounter plugins in ImageJ.

**Analysis of cell shape and alignment**. ImageJ analysis software was used to measure cell alignment and cell elongation. Alignment was determined as the angle between the long axis of the cell and the horizontal axis of the image at 180º and values were expressed as the percentage of cells at a particular angle of orientation; Elongation was determined as 1- circularity [long axis/short axis of the cell].

**Western blotting**. Detailed procedures of cell lysis, electrophoresis and western blotting are provided in Supplemental Methods. Primary and secondary antibodies used for western blotting are listed in Supplementary Table 12. Band intensity was determined by densitometry using ImageJ software. Expression of proteins of interest was normalised to β-actin.

**Statistics and reproducibility**. All graphs were either plotted in RStudio or Graphpad Prism 8 software (Graphpad Software Inc, CA, USA), with statistical tests performed, as appropriate. In bar graphs, error bars indicate the standard error of the mean (SEM). Source data for graphs are in Supplementary Data 17. When comparing two sample groups, normally distributed data were analysed using an unpaired Student's t-test. For comparisons of two or more sample groups, one or two-way ANOVA was used, as appropriate. The threshold for statistical significance was $P < 0.05$.

**Study approval**. Venous blood samples were obtained from local ethics committee approval and informed written consent from healthy volunteers and idiopathic PAH patients with ethics committee approval and informed written consent (REC Ref 17/LO/0563). Participants were identified by number.

**Reporting summary**. Further information on research design is available in the Nature Research Reporting Summary linked to this article.

## Data availability

All data generated or analysed during this study are included in this published article and its supplementary information files, except for RNA sequencing data, which cannot be shared publicly because of the sensitive nature and risk of identifying patients. Data are available from the Imperial College London Institutional Data Access/Ethics Committee (contact via Ms. Becky Ward, Research Governance Manager, E: becky.ward@imperial.ac.uk) for researchers who meet the criteria for access to confidential data. Any other information is available from the corresponding author upon request.

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

## Acknowledgements

This research was supported by the BHF PhD Studentship FS/17/64/33476 (Alex Ainscough), British Heart Foundation project grant PG/19/19/34286, BHF Imperial Centre of Research Excellence grant RE/18/4/34215 and ICIC (MRC) Confidence in Concept 20/21 grant 4050789012. J.B.E. has been supported by the European Research Council

(ERC) under the European Union's Horizon 2020 research and innovation programme (grant agreement Nos. 724300 and 875525). We thank the staff of the NIHR Imperial Clinical Research Facility, Hammersmith Hospital (London UK), Mr. Lee Tooley of the Imperial College Mechanical Instrumentation Workshop for his help in fabricating the racks and trays for the flow system, Mr. Joshua Dodd (Morgan Branding) for help in the preparation of scientific illustrations, Dr. Hebah Sindi (Imperial College London) for help with immunostaining of human lung sections and Dr. Sandro Satta (Imperial College London) for help with optimising RNA extractions and qPCR. We are particularly grateful to Mr. Steve Rothery (Imperial FILM facility) for his help and advice with image acquisition and analysis. All widefield/confocal imaging analysis was performed at the Facility for Imaging by Light Microscopy (FILM) at Imperial College London, which is part-supported by funding from the Wellcome Trust (grant 104931/Z/14/Z) and BBSRC (grant BB/L015129/1).

## Author contributions

A.J.A. performed chip fabrication, designed and performed experiments, analysed data and wrote the manuscript; T.S. performed in silico computational simulations, and wrote the manuscript; C.J.R. provided code and assisted with performing RNAseq analysis and critically evaluated the manuscript; E.V. performed SOX17 binding sites analysis; S.S. and A.F performed qPCRs; MRW, JW & LSH provided patient material and critically evaluated the manuscript; K.G. and A.F. provided drug compounds supported drug testing study; H.W. carried out proteomic analysis; P.D.U. and B.D. provided BMPR2 mutant mouse lung tissues; M.H. performed chip fabrication and experiments; J.B.E. provided guidance on in silico simulations, chip design and fabrication, secured funding and critically evaluated the manuscript. BWS conceived the study, designed experiments, secured funding and wrote the manuscript.

## Competing interests

Ambrisentan and AZD5153 used in this study were provided by Astra Zeneca.
