## [Peer Review File · Communications Biology]

Reviewers' comments:

Reviewer #1 (Remarks to the Author):

This study is about converting PAECs and PSMCs into PAH-afflicted pulmonary arterial endothelial (PAH-ECs) and smooth muscle cells (PSMCs) by means of BMPR2 knockdown and hypoxia. The investigators used a vertical microfluidic device to recapitulate PAH-afflicted pulmonary arteries. The authors also collected ECFC and let them differentiate into ECs and apparently used again BMPR2 knockdown and hypoxia to develop PAH-like PAECs. The team generated a large amount of data in support of their hypothesis that upon BMPR2 knockdown and hypoxic treatment, PAECs, ECFC and PSMC show PAH like phenotypes and genotypes. While the authors made a convincing case that ECs and SMCs acquire PAH like cellular features upon BMPR2 knockdown and hypoxic treatment, but their claim that the device can be a model of PAH-afflicted pulmonary arteries is a huge overstatement. Below is a list of issues the authors overlooked when claiming a two-layer vertical device as a pulmonary artery.

1. An artery must have three layers of cells as the author nicely presented in Fig. 2a. In the absence of adventitial layer, we cannot claim that a two-layer device recapitulate artery like anatomy or physiology.
2. One of the major pathologies of PAH is that misplaced cell growth, migration of ECs to SMCs layer or to adventitial layer. The cells separated by PET of 400 nm pore cannot move from EC layer to SMC layer, given ECs are typically 10-30 μ wide. The presence of PET may allow cell-cell communication via molecular cues, but not direct cell-cell communication such as paracrine signaling.
3. It is unclear whether the authors studied the influence of PAH-like ECs on healthy SMCs or that of PAH-like SMCs on healthy ECs, given endothelial-mesenchymal transition is common in PAH.
4. Novelty of the device is also questionable because similar devices have been used for emulating many tissue-like functions.
5. Statements that the device can be used for drug screening or for studying PAH pathophysiology is oversold because the authors only focused on one type of PAH, when PAH has many subtypes.
6. Suggest that the authors focus on converting ECs and SMCs to PAH-like features instead of focusing on the device. Refocusing the work on developing cellular models of PAH would make more sense instead of focusing on the device. The authors deserve high praise because of the huge amount of data presented in support of the cellular models.
7. Suggest removing "microfluidic model" to "novel model" and thus change the discussion and introduction accordingly.

Overall, the authors put a tremendous amount of work in putting together the manuscript, which merits attention by the bigger scientific community.

Reviewer #2 (Remarks to the Author):

This is a very original and interesting study on the establishment of a PAH on the chip model. The authors created a couple of 2-hit models combining BMPR2 deficiency with hypoxia in either HPAEC/HPASMC or Ecfc-PAH/HPASMC. These models showed that the proposed model recapitulates many features seen in PAH patients. In addition, they showed that this unique model of SOX17 upregulation can prevent HPASMC proliferation through the prostacyclin signaling pathway and that ETRA or BRD4 inhibitors could improve the PAH phenotype in their model.

Overall, I found the study very observational in nature (proof of concept study) and heavily relying on RNAseq. Although confirmed by qPCR the main target could have also been confirmed by western blotting.

Authors should also explore effects on apoptosis and metabolism and not only on proliferation. To determine if their model could also be used to test drugs on other feature of PAH and not only proliferation.

Authors should also acknowledge limitations of the model and compare it to animal models. In fact many reviews have highlighted limitation of the animal models in PAH but also have highlighted that a better utilization of the model could also improve PAH research. Please discuss the following
<https://doi.org/10.1161/CIRCRESAHA.117.312579>
<https://doi.org/10.1161/CIRCRESAHA.121.319971>

In addition how is calculated sample size in your system. Do we need to build 5-10 chips with different patients cells ? this should also be discussed especially since utilization of human cells has also some limits please discussed <https://doi.org/10.1164/rccm.201607-1515PP>

Finally, how the authors see translation potential of findings on PAH chips to human clinics i.e. how can toxicity and so on can be assessed ? Do you think that your system will be the end of animal studies ?

How your system can be compare to culture human lungs slice ?

Thank you

Reviewer #3 (Remarks to the Author):

The Authors, in a multicenter collaboration including an industry partner, present a novel model system for pulmonary arterial hypertension research. They go on to test the model system using a few different approaches (exposure (e.g., hypoxia) and genetic (e.g., BMPR2)). They argue that the The model identifies relevant alterations in the pulmonary endothelial cell phenotype relevant to smooth muscle cell remodelling believed to occur in PAH tissues. that are essential for the induction of SMC remodelling. They also identify relationships at the molecular level perhaps relevant to hPAH and other PAH forms.

Major Comments, Introduction:

A) Line 50: "Animal models do not fully reproduce the features of human PAH, which is a key obstacle to drug development.8"

This sentence is somewhat misleading and in the eye of the beholder. While most do not (almost all do not), there are some emerged genetic models which are quite close. The Reference #8, while a

nice manuscript, does not even mention BMPR2 or the BMP pathway, which is rather odd considering it is by far the most relevant genetic variation in human PAH and odd that the murine BMPR2 models are quite similar. Now rodents are not perfect, and have many other challenges for which novel in vitro model systems are highly relevant so no quibble with the concept--just not sure the 'sell' here on the in vivo models was completely balanced. Would suggest adjustment.

Major Comments, Methods:

A) A substantial potential challenge is populating these types of model systems with cells that truly recapitulate 'pulmonary' cells. The use of ECs (HPAECs) from Promo cell would benefit from more information: what size vessels are these derived from? How many passages (presumably a lot?), etc... Also, from what type of person are the Lonzo HPASMC derived? Are these PAH patients, or are they 'normals' who died of trauma, etc...?

B) Transfection efficiencies were assessed for knockdowns, but do we know how long and/or passages they last?

C) It is noted that "blood-derived endothelial colony-forming cells (ECFCs) are often used as surrogates for pulmonary endothelial cells in PAH and display abnormalities in key pathways linked to the disease pathogenesis^{19,20}" This is true at the study level such as in these two refs, but those are very different studies relative to this chip study. Are there other refs and/or assurances that these derived ECs are truly like pulmonary ECs, and at what sizes of the vasculature, etc...?

Major Comments, Results:

A) Excellent figures, thank you for that effort for the readers. Same for extra associated files. Similar to the question above, how 'durable' over time is the BMPR2 knockdown (2c) and how relevant is this to the model understanding of the true vascular condition? Relevant to 2d and Suppl S6.

B) The data on concurrent combination of BMPR2 knockdown with hypoxia is interesting. But, big picture not totally relevant to the model itself but to its data in this manuscript: how relevant is this to the 'human condition'? That is, iPAH and hPAH patients are not typically 'hypoxic'.

Response to the Reviewers' comments:

We are grateful to the Reviewers for their helpful comments. Our responses are marked in red.

Reviewer #1 (Remarks to the Author):

This study is about converting PAECs and PSMCs into PAH-afflicted pulmonary arterial endothelial (PAH-ECs) and smooth muscle cells (PASMCS) by means of BMPR2 knockdown and hypoxia. The investigators used a vertical microfluidic device to recapitulate PAH-afflicted pulmonary arteries. The authors also collected ECFC and let them differentiate into ECs and apparently used again BMPR2 knockdown and hypoxia to develop PAH-like PAECs. The team generated a large amount of data in support of their hypothesis that upon BMPR2 knockdown and hypoxic treatment, PAECs, ECFC and PSMC show PAH-like phenotypes and genotypes. While the authors made a convincing case that ECs and SMCs acquire PAH-like cellular features upon BMPR2 knockdown and hypoxic treatment, but their claim that the device can be a model of PAH-afflicted pulmonary arteries is a huge overstatement. Below is a list of issues the author overlooked when claiming a two-layer vertical device as a pulmonary artery.

1. An artery must have three layers of cells as the author nicely presented in Fig. 2a. In the absence of adventitial layer, we cannot claim that a two-layer device recapitulate artery-like anatomy or physiology.

Response: Our device models intimal-medial cell interactions in small pre-capillary arterioles, where these two layers predominate. In our approach, we aimed to reproduce PAH responses in a two-cell layer system (in particular medial cell proliferation) to facilitate quick evaluation of drug effects. This is a reductionist model which, like other organ-on-a-chip models, does not fully replicate the structural and functional complexity of human organs.

To explain the purpose of our device more clearly, we have replaced the term “model of live vessel” with “model of arteriolar cell-cell interactions” and, in another sentence, the phrase “... was designed to monitor responses of human pulmonary vascular cells to pathological and potential therapeutic interventions”, with “... was designed to monitor responses of human pulmonary vascular endothelial and smooth muscle cells to pathological and potential therapeutic interventions” (Introduction/Results page 3, lines 60 and 73).

2. One of the major pathologies of PAH is that misplaced cell growth, migration of ECs to SMCs layer or to adventitial layer. The cells separated by PET of 400 nm pore cannot move from EC layer to SMC layer, given ECs are typically 10-30 μ wide. The presence of PET may allow cell-cell communication via molecular cues, but not direct cell-cell communication such as paracrine signaling.

Response: Cells grown on either side of the porous PET membranes with 0.4 μ m pore size can communicate directly by exchanging signalling molecules between the two compartments (doi.org/10.1016/j.engreg.2022.05.001). The membrane accommodates passage of FITC-dextran MW 40kDa, which has a Stokes radius of serum albumin (doi.org/10.1161/01.RES.76.2.199), from the endothelial to the SMC compartment. 0.4 μ m pore size also allows gap junctional communication through membrane protrusions extended through the pores but, as pointed out by the Reviewer, does not allow cell migration. The design can be modified by introducing a membrane with larger pores (8-10 μ m) and we plan to test this response in our future studies.

3. It is unclear whether the authors studied the influence PAH like ECs on healthy SMCs or that of PAH like SMCs on healthy ECs, given endothelial mesenchymal transition is common in PAH.

Response: We studied the influence of PAH-like endothelial cells on SMC phenotype by culturing healthy HPASMCs with endothelial cells with natural or induced BMPR2 deficiency, combined with hypoxia.

4. Novelty of the device is also questionable because similar device has been used for emulating many tissue-like functions.

Response: We mentioned this in Results on page 4, lines 82 and 83

("...The basic principles of the design were derived from the lung-on-a-chip model").

5. Statements that the device can be used for drug screening or for studying PAH pathophysiology is oversold because the authors only focused on one type of PAH, when PAH has many subtypes.

Response: Medial cell hyperplasia is a common feature of all sub-groups of PAH and, although a complete picture of PAH pathogenesis is not fully understood, hypoxia together with the loss of BMPR2 function (caused by genetic mutations, protein degradation or reduced gene expression: doi: 10.3390/ijms19092499), are regarded as important contributory factors. While these two factors may not be the only ones involved in the disease pathogenesis, they were sufficient to induce PASMC proliferation. In addition, the set of differentially expressed genes showed a substantial overlap with the known PAH databases.

We have modified the final sentence of Discussion (page 14, line 360) "...Significant overlaps with pre-existing lists of differentially expressed PAH genes, are amenable for novel target discovery/validation and can potentially be applied for use in drug screening or toxicology".

6. Suggest that the authors focus on converting ECs and SMCs to PAH-like features instead of focusing on the device. Refocusing the work on developing a cellular models of PAH would make more sense instead of focusing on the device. The authors deserve high praise because of the huge amount of data presented in support of the cellular models.

Response: It was important to us to describe the device fabrication, optimization and validation in detail, so it can be reproduced by others. This was exactly the stumbling block when we started this project – most publications, including the renowned Nature articles, did not contain sufficient detail on how to fabricate and validate these devices so we had to spend a long time optimising the protocols.

Therefore, if it is acceptable by the Reviewer and the Editors, we would like to retain the current format of the manuscript.

7. Suggest removing "microfluidic model" to "novel model" and thus change the discussion and introduction accordingly.

Response: We wanted to emphasize that this is an organ on a chip model of PAH. However, as the "PAH-on-a-chip" title had already been used in another publication (<https://pubs.rsc.org/en/content/articlelanding/2020/lc/d0lc00605j>), we used the term "microfluidic model" to indicate that the work utilises a miniaturised device handling small amounts of fluid. The phrase "microfluidic model of..(organ or disease)" has also been used by others (DOI: 10.1007/s12195-016-0469-0; doi.org/10.1038/srep36086; DOI: 10.1126/scitranslmed.aba902).

We suggest alternative titles “A microengineered novel model of pulmonary arterial hypertension” or “An Organ-on-a-chip model of endothelial-smooth muscle interactions in pulmonary arterial hypertension uncovers a BMPR2-SOX17-prostacyclin signalling axis”. We would be grateful for the Reviewer’s and Editors suggestions of the most appropriate choice.

Overall, the authors put a tremendous amount of work in putting together the manuscript, which merits attention by the bigger scientific community.

Reviewer #2 (Remarks to the Author):

This is a very original and interesting study on the establishment of a PAH on the chip model.

The authors created a couple of 2 hits models combining BMPR2 deficiency with hypoxia in either HPAEC/HPASMC or Efc-PAH/HPASMC showed that the proposed model recapitulate many features seen in PAH patients. In addition they showed that this unique model SOX17 upregulation can prevent HPASMC proliferation through prostacyclin signalling pathway and that ETRA or BRD4 inhibitors could improve PAH phenotype in their model.

Overall, I found the study very observational in nature (proof of concept study) and heavily relying on RNAseq. Although confirmed by qPCR the main target could have also been confirmed by western blotting.

Response: We have carried out an additional analysis of SOX17, VE-cadherin and KCNK1 expression changes in HPAECs and HPASMCs by immunostaining (please see Results page 6 line 148-152 and new Supplementary Figure 12 in the revised manuscript).

Our chip contains only ~ 8,000 cells/channel and therefore can not provide sufficient material for protein detection by western blotting. Preliminary experiments showed that at least ~ 50,000 HPAECs would be required for the detection of β -actin by western blotting. This agrees with a theoretical evaluation of the number of cells required to generate 10ug protein for analysis, taking into account that a single mammalian cell contains ~ 0.2-0.25ng protein (Bionumbers database*) Owing to the time and cost of chip fabrication, the only feasible methods accessible to us were qPCR and the semi-quantitative analysis of immunofluorescence. Development of proteomic workflows is of considerable interest and we are currently working on optimising cell extraction procedures for use in quantitative proteomics.

*Reference: (<https://bionumbers.hms.harvard.edu/> ;<https://bionumbers.hms.harvard.edu/bionumber.aspx?id=110558&ver=1&trm=protein+content+per+cell&org=> ; Milo R; Jorgensen P; Moran U; Weber G; Springer M, BioNumbers—the database of key numbers in molecular and cell biology. Nucleic Acids Res. 2009, 38 (suppl_1), D750–D753.)

Figure S12. Expression changes of selected protein targets in HPAECs and HPASMCs in the “double hit” microfluidic PAH model. (A) and (B) are representative confocal images of fluorescently labelled SOX17, VE-cadherin in HPAECs and KCNK1 in HPASMCs, treated, as indicated. Bar=10 μ m. (C, D, E) are corresponding graphs showing changes in protein expression of these targets in normoxic controls (Normoxia Adcontrol) and cells under the “double hit” conditions (Hypoxia BMPR2 shRNA). Semi-quantitative analysis of fluorescence intensity, * $P < 0.05$, ** $P < 0.01$, comparison with normoxic Adcontrol. Error bars indicate mean \pm SEM, unpaired t-test, $n=5$.

Authors should also explore effects on apoptosis and metabolism and not only on proliferation. To determine if their model could also be used to test drugs on other feature of PAH and not only proliferation.

Response: In addition to the analysis of cell apoptosis in the ECFCs shown in the original manuscript (marked as Fig. S15 in the revised manuscript), we have carried out analysis of cell apoptosis in the inducible HPAEC “double hit” model (please see new Figure S7 in the revised manuscript and Results page 5, lines 111-112). Both analyses showed no effect on endothelial or SMC apoptosis in cells cultured under the “double hit” conditions for 24h. However, we can not exclude a possibility that a longer exposure would reveal a more profound effect and plan to observe cell responses longer-term culture in our future studies.

In our current device, 8-10,000 cells are perfused with 3ml of fluid which, considering the large dilution factor, precludes the measurement of circulating metabolites by H-NMR spectroscopy. We are currently optimising the device to reduce the volume of circulating medium.

Figure S7. Effect of hypoxia and BMPR2 knockdown on endothelial and smooth muscle cell apoptosis in the HPAEC “double hit” model. (A) Representative confocal images of HPAEC channel, with apoptotic cell nuclei labelled blue (Click-iT Plus TUNEL Alexa Fluor 647) and endothelial cell nuclei labelled green (immunofluorescence, anti-erg antibody) in cells treated, as indicated. Cells treated with DNase I were used as positive control. Arrowheads point to apoptotic endothelial cells. Bar=100µm. (B) Graph showing % of apoptotic cells in HPAEC and HPASMC channel. Ns-non-significant, unpaired Student t-test, n=4. Apoptotic HPAECs were identified by blue/green erg-positive nuclei, while apoptotic HPASMCs were marked by blue nuclei that were erg-negative (blue only).

As a proof of concept and to validate the observations derived from RNAseq analysis, we measured the activity of hexokinase, the rate-limiting enzyme providing glucose 6-phosphate for glycolysis and pentose cycle in PSMCs (please see new Supplementary Figure S13, Results page 6 line 149-153, Discussion page 12, lines 301-305). The results showed a marked activation of hexokinase under hypoxia and, what is most interesting, a further significant activation of the enzyme under the “double hit” conditions. This was consistent with the RNAseq results showing an upregulation of HKII expression in PSMCs, as well as their heightened proliferative potential, which is highly reliant on the energy and metabolites from glycolysis and the pentose cycle.

Figure S13. Hexokinase activity in PASCs under normoxic, hypoxic and “double hit” conditions.

Hexokinase activity was measured in pulled samples of PASCs (2×10^5 cells/treatment) co-cultured with HPAECs under normoxic conditions (Adcontrol Normoxia), under hypoxia (Adcontrol Hypoxia) and under the “double hit” conditions (AdBMPR2 shRNA Hypoxia). Hexokinase activity under basal, control conditions was 0.86 ± 0.043 nM/min/mL. * $P < 0.05$; *** $P < 0.0001$, comparison with normoxic Adcontrol; # $P < 0.05$, comparison, as indicated. Error bars indicate mean \pm SEM of a one-way ANOVA with a Tukey’s post-hoc correction test, $n=5$.

Authors should also acknowledge limitations of the model and compare it to animal models. In fact many reviews have highlighted limitation of the animal models in PAH but also have highlighted that a better utilization of the model could also improve PAH research. Please discuss the following <https://doi.org/10.1161/CIRCRESAHA.117.312579>

<https://doi.org/10.1161/CIRCRESAHA.121.319971>

Response: We have included additional comments regarding benefits and limitations of animal models as well as limitations of our model in Discussion, pages 13 and 14 lines 348-359.

In addition how is calculated sample size in your system. Do we need to build 5-10 chips with different patients cells ? this should also be discussed especially since utilization of human cells has also some limits please discussed <https://doi.org/10.1164/rccm.201607-1515PP>

Response: The sample size would vary, depending on the assay. In permeability or proliferation studies, the use of cells from 3 different biological donors with 4-5 experimental repeats were sufficient to detect significant differences and the sample size was based on published reports using these two well established assays. RNAseq involved the analysis of cells from 4-5 different biological donors with each donor having two technical repeats (2 chips/donor) which were pooled together in order to reduce inter-chip variability and to ensure that a surplus amount of RNA could be extracted. For any new assay performed using the chip we would suggest running a minimum of 3 biological samples with 2 technical replicates (chips) prior to performing power calculations to determine the final number of chips required.

Limitations of our model and the need of stringent study design are mentioned in Discussion p13 and 14, lines 348-359:

“...Animal models have been instrumental in deciphering the molecular mechanisms underlying the disease but are often criticised for not fully reproducing the pathology of human disease and low success in clinical translation of new drugs⁸. Benefits and limitations of animal models of PAH as well as new measures undertaken to improve their translational value have been summarised in the two excellent recent reviews^{8,54,55}. Our model successfully reproduced the disease characteristics previously described in human and animal PAH, such as medial cell proliferation and activation of PAH-associated pathways in pulmonary arterial intimal and medial cells. However, considering the limitations of this model, such as lack of complexity, variability associated with donor genetic diversity, age and comorbidities⁵⁵, as well as potential differences in manufacturing protocols amongst laboratories, attention to stringent study design⁵⁵, including data validation with robust animal models, may be required in future device applications in exploratory and confirmatory PAH preclinical studies. ...”

Finally, how the authors see translation potential of findings on PAH chips to human clinics i.e. how can toxicity and so on can be assessed ? Do you think that your system will be the end of animal studies ?

Response: From the translational point of view, future application of isogenic patient-derived cells in the device is the most exciting prospect, as it would allow testing of patient-specific responses to drug treatment (cell viability/apoptosis, proliferation, barrier function, changes in PAH-associated gene expression). Considering the existing limitations of this model (and other organs-on-chips), we would recommend the use of a single robust animal model for data validation (Discussion p. 13 and 14 lines 348-359.).

Investigating the effects of specific mutations using cells from HPAH patients would help understand some of the pathological mechanisms in PAH but, considering a limited access to such cells, the work would have to be done cooperatively, perhaps in selected dedicated laboratories.

Our inducible model offers a solution by defining conditions needed to achieve some of the key characteristics of PAH endothelial and medial cells, which can be reproduced in other laboratories. We also use patient blood-derived endothelial cells, which are far more accessible than lung explants.

How your system can be compare to culture human lungs slice ?

Response: Lung slices are an important tool in aiding the discovery process or helping data validation but require a high level of technical skills and access to the source material (lung explants), which, as opposed to blood-derived cells, is extremely limited, so it is difficult to imagine how they could be used as platforms for the evaluation of drug efficacy or toxicity.

Reviewer #3 (Remarks to the Author):

The Authors, in a multicenter collaboration including an industry partner, present a novel model system for pulmonary arterial hypertension research. They go on to test the model system using a few different approaches (exposure (e.g., hypoxia) and genetic (e.g., BMPR2). They argue that the The model identifies relevant alterations in the pulmonary endothelial cell phenotype relevant to smooth muscle cell remodelling believed to occur in PAH tissues. that are essential for the induction

of SMC remodelling. They also identify relationships at the molecular level perhaps relevant to hPAH and other PAH forms.

Major Comments, Introduction:

A) Line 50: "Animal models do not fully reproduce the features of human PAH, which is a key obstacle to drug development.⁸"

This sentence is somewhat misleading and in the eye of the beholder. While most do not (almost all do not), there are some emerged genetic models which are quite close. The Reference #8, while a nice manuscript, does not even mention BMPR2 or the BMP pathway, which is rather odd considering it is by far the most relevant genetic variation in human PAH and odd that the murine BMPR2 models are quite similar. Now rodents are not perfect, and have many other challenges for which novel in vitro model systems are highly relevant so no quibble with the concept--just not sure the 'sell' here on the in vivo models was completely balanced. Would suggest adjustment.

Response: We have provided a new reference (doi.org/10.1161/CIRCRESAHA.121.319971) that discusses benefits and limitations of currently used animal models in PAH.

In Discussion on page 13 and 14 lines 348-357, we now explain that organ on a chip models have some important limitations and while animals models have been criticised, they remain a key research tool in investigating pathophysiology of PAH and that the use of at least one robust animal model would be recommended in validation of data obtained with OOaC modelling.

Major Comments, Methods:

A) A substantial potential challenge is populating these types of model systems with cells that truly recapitulate 'pulmonary' cells. The use of ECs (HPAECs) from Promo cell would benefit from more information: what size vessels are these derived from? How many passages (presumably a lot?), etc... Also, from what type of person are the Lonzo HPASMC derived? Are these PAH patients, or are they 'normals' who died of trauma, etc...?

Response: New Supplementary Table S2 on page 14 of Supplement provides information about the lot number, sex, age and population doubling time of cells used in our study. These were macrovascular pulmonary arterial endothelial and smooth muscle cells and donors were non-smokers. We do not know the specific isolation protocols applied by Promocell and Lonza but believe that the cells were derived from large arteries as well as arterioles. Cells were acquired at passage 3 and used at passages 5-6. This information has now been provided in Methods on page 15, lines 387-388.

B) Transfection efficiencies were assessed for knockdowns, but do we know how long and/or passages they last?

Response: Our experiments were performed within 48h of adenoviral infection on 90% confluent cells. Cells infected with adenoviruses continue to express the gene of interest until they become apoptotic/die. As adenoviral plasmids are not integrated with the cell's genome and do not replicate, their effect eventually becomes "diluted" during subsequent passages. This was not the case in our study, as we investigated the behaviour of confluent endothelial cells within a relatively space of time.

C) It is noted that "blood-derived endothelial colony-forming cells (ECFCs) are often used as surrogates for pulmonary endothelial cells in PAH and display abnormalities in key pathways linked to the disease pathogenesis^{19,20}" This is true at the study level such as in these two refs, but those are very different studies relative to this chip study. Are there other refs and/or assurances that these derived ECs are truly like pulmonary ECs, and at what sizes of the vasculature, etc...?

Response: ECFC have been used to investigate endothelial molecular dysfunction in several diseases, as they give access to endothelial cells from patients in a non-invasive way.

ECFCs have unique characteristics, for instance they express markers indicating a myeloid stem cell origin (CD34, c-kit) and resemble endothelial cells in morphology and EC marker expression (i.e., CD144, CD31, VEGFR-2, VWF) (doi.org/10.3389/fmed.2018.00295; [doi: 10.3390/ijms19123763](https://doi.org/10.3390/ijms19123763)).

Major Comments, Results:

A) Excellent figures, thank you for that effort for the readers. Same for extra associated files. Similar to the question above, how 'durable' over time is the BMPR2 knockdown (2c) and how relevant is this to the model understanding of the true vascular condition? Relevant to 2d and Suppl S6.

Response: The effect of adenovirus-mediated knockdown can be observed for at least a week (in a quiescent, growth-inhibited cell monolayer for as long as the cells stay alive). In our model we focused on early endothelial and smooth muscle responses to vascular insults and the transcriptomic "phenotype" suggests that our "two hit" model of PAH offers a snapshot of endothelial dysfunction seen in early disease, associated with a loss of arterial identity, reduced eNOS signalling and increased susceptibility to damage, accompanied by inflammation and metabolic shift towards glycolysis (Discussion on page 11 lines 288-291).

We are currently working on extending cell culture time and modifying the device to accommodate measurement of endothelial proliferation and angiogenesis to facilitate modelling of cell behaviour seen in the end-stage human PAH.

B) The data on concurrent combination of BMPR2 knockdown with hypoxia is interesting. But, big picture not totally relevant to the model itself but to its data in this manuscript: how relevant is this to the 'human condition'? That is, iPAH and hPAH patients are not typically 'hypoxic'.

Response: Hypoxia and BMPR2 loss of function are known to contribute to pulmonary vascular remodelling in human PAH and pre-clinical models of the disease. However, the exact contribution of each of these factors is not known and we hope that our study, which identified subsets of differentially expressed genes induced by hypoxia and BMPR2 knockdown applied alone or in combination, will shed some light on this issue.

There are likely to be many more factors required, in addition to hypoxia and BMPR2, to produce the full disease phenotype. The relevance of our model to human condition however is illustrated by an overlap between our and other known transcriptomic PAH databases as well as medial SMC hyperplasia, which is a shared characteristics of vascular phenotype in PH.

Reviewers' comments:

Reviewer #1 (Remarks to the Author):

The authors did not even make a few editorial corrections given that this model does not fully reflect the neither the biology nor the pathophysiology of PAH afflicted pulmonary arteries.

Only item the authors chose or suggested to change is the change of title "An Organ-on-a-chip model of endothelial-smooth muscle interactions in pulmonary artery hypertension uncovers a BMPR2-SOX17-prostacyclin signalling axis", which sounds reasonable, but the author should remove the word "hypertension".

This reviewer continues to believe that the statement "this model can emulate pulmonary arterial hypertension" is a huge overstatement.

Use of model for pulmonary arterial cell-cell interaction is accurate. When the authors agree that this model does not allow cellular migration, a hallmark of PAH, then how this model can be claimed as a model for pulmonary hypertension.

In addition to the paper cited by the author (Al-Hilal et al), a protocol paper detailing the fabrication of PAH chip model has been published this month (Miromachines, 2022 Sep 7;13(9):1483. doi: 10.3390/mi1309148). This protocol paper also explains how multichannel chip model is far superior than vertical models separated by a membrane.

The reviewer also suggests that the authors discuss the limitations of this model when compared with the published model or vice versa.

The reviewer applauds the amount of work the authors put in this manuscript and thus strongly suggest to make those editorial which does not require any new experiment just correction of few phrases and the title.

Reviewer #2 (Remarks to the Author):

No further comments

Reviewer #3 (Remarks to the Author):

I appreciate the thoughtful replies to each Reviewer in the 'response to reviewers', and the alterations to the manuscript.

I would suggest the following title, but certainly defer to the Editorial Team:
"A novel microengineered model of pulmonary arterial hypertension"

Response to Reviewer's comments

Reviewer: The authors did not even make a few editorial corrections given that this model does not fully reflect the neither the biology nor the pathophysiology of PAH afflicted pulmonary arteries...

Response: It was not our intention to claim that the model "fully reflects" or "(fully) emulates" the biology or the pathophysiology of PAH-afflicted pulmonary arteries.

To clarify this further, we have made the following changes:

On page 3 lines 59-61 we explain that this work presents "...a microfluidic model of human pulmonary arterial endothelial-smooth muscle cell interactions cultured under the conditions of BMPR2 knockdown and hypoxia, the two known triggers of PAH. In this manuscript, this model is referred to as a "two hit" model of PAH".

We have replaced the expression "a model of PAH" with "a model of pulmonary arterial endothelial-smooth muscle cell interactions in PAH" in the Abstract on page 2, line 23 and on page 3, line 59-60 and in Discussion on page 10 line 256.

The reason behind retaining a reference to PAH in this phrase and in the new title is that our model was constructed with the view of studying PAH-associated processes: we used two known triggers of PAH (hypoxia and natural and induced BMPR2 deficiency), cells from PAH patients, validated results with PAH databases and used approved PAH therapeutics.

In Discussion on p.11 we say that the model "offers an insight to processes seen during development of the disease.." (instead of "reflects" or "emulates")

In the final paragraph of Discussion, on page 14, line 462, "in PAH" was removed. We now say: "In summary, we are first to provide a microfluidic and inducible model of vascular cell dysfunction, informing of functional and transcriptomic effects of endothelial BMPR2 deficiency and hypoxia. Significant overlaps with pre-existing lists of differentially expressed PAH genes, are amenable for novel target discovery/validation and can potentially be applied for use in drug screening or toxicology.

Reviewer: Only item the authors chose or suggested to change is the change of title "An Organ-on-a-chip model of endothelial-smooth muscle interactions in pulmonary artery hypertension uncovers a BMPR2-SOX17-prostacyclin signalling axis", which sounds reasonable, but the author should remove the word "hypertension".

Response: By definition, "a model" presents a simplified image and in this context, a reference to PAH should not be taken as an overstatement. As Norbert Wiener said, "The best material model of a cat is another, or preferably the same, cat." but such an approximation is not achievable.

Our model was constructed with the view of studying PAH-associated processes: we used two known triggers of PAH (hypoxia and natural and induced BMPR2 deficiency), cells from PAH patients, validated results with PAH databases and used approved PAH therapeutics and therefore including a reference to pulmonary hypertension in the title is, in our view, appropriate.

"Pulmonary hypertension" appears in the titles of other publications presenting microfluidic models (<https://doi.org:10.3390/mi13091483>; <https://doi.org:10.1039/d0lc00605j>)

This reviewer continues to believe that the statement "this model can emulate pulmonary arterial hypertension" is a huge overstatement.

Response: We have modified the potentially misleading phrases (please see our response above) and do not claim to fully emulate pulmonary arterial hypertension.

Reviewer: Use of model for pulmonary arterial cell-cell interaction is accurate. When the authors agree that this model does not allow cellular migration, a hallmark of PAH, then how this model can be claimed as a model for pulmonary hypertension. In addition to the paper cited by the author (Al-Hilal et al), a protocol paper detailing the fabrication of PAH chip model has been published this month (Miromachines, 2022 Sep 7;13(9):1483. doi: 10.3390/mi1309148). This protocol paper also explains how multichannel chip model is far superior than vertical models separated by a membrane. The reviewer also suggests that the authors discuss the limitations of this model when compared with the published model or vice versa.

Response: In Discussion on page 13, lines 344-350 we discuss limitations of our model : "...Our device, which accommodates only 2 cell types, is unlikely to fully reflect complex pathophysiology of PAH and therefore future sourcing of patient-derived cells and introducing other cell types will be a vital consideration. The current design should be further improved to allow migration of cells between the two vascular layers, a response linked to vascular remodelling in PAH. One such approach would be increasing the size of pores in the membrane separating the two microfluidic channels..."

The title of the paper (Miromachines, 2022 Sep 7;13(9):1483. doi: 10.3390/mi1309148) is : "A Protocol for Fabrication and on-Chip Cell Culture to Recreate PAH-Afflicted Pulmonary Artery on a Microfluidic Device" which, applying the same logic, promises "a PAH-afflicted artery on a microfluidic device". The authors also claim that their device "... emulates the major clinical features of PAH, such as arterial remodeling, muscularization, plexiform lesions, and sex disparity", which, in the context of Reviewer's comments, looks like an overstatement.

We have included this reference for those who wish to learn more about the fabrication and application of Hilal's model.

The reviewer applauds the amount of work the authors put in this manuscript and thus strongly suggest to make those editorial which does not require any new experiment just correction of few phrases and the title.

We thank the Reviewer for reviewing the manuscript and hope that they find our responses satisfactory.